# Improving Convergence and Generalization Using Parameter Symmetries

**Bo Zhao**
University of California San Diego
bozhao@ucsd.edu

**Robert M. Gower**
Flatiron Institute
rgower@flatironinstitute.org

**Robin Walters**
Northeastern University
r.walters@northeastern.edu

**Rose Yu**
University of California San Diego
roseyu@ucsd.edu

## Abstract

In many neural networks, different values of the parameters may result in the same loss value. Parameter space symmetries are loss-invariant transformations that change the model parameters. Teleportation applies such transformations to accelerate optimization. However, the exact mechanism behind this algorithm's success is not well understood. In this paper, we show that teleportation not only speeds up optimization in the short-term, but gives overall faster time to convergence. Additionally, teleporting to minima with different curvatures improves generalization, which suggests a connection between the curvature of the minimum and generalization ability. Finally, we show that integrating teleportation into a wide range of optimization algorithms and optimization-based meta-learning improves convergence. Our results showcase the versatility of teleportation and demonstrate the potential of incorporating symmetry in optimization.

## 1 Introduction

Given a deep neural network architecture and a dataset, there may be multiple points in the parameter space that correspond to the same loss value. Despite having the same loss, the gradients and learning dynamics originating from these points can be very different (Kunin et al., 2021; Van Laarhoven, 2017; Grigsby et al., 2022). Parameter space symmetries, which are transformations of the parameters that leave the loss function invariant, allow us to *teleport* between points in the parameter space on the same level set of the loss function (Armenta et al., 2023). In particular, teleporting to a steeper point in the loss landscape leads to faster optimization.

Despite the empirical evidence, the exact mechanism of how teleportation improves convergence in optimizing non-convex objectives remains elusive. Previous work shows that gradient increases momentarily after a teleportation, but could not show that this results in overall faster convergence (Zhao et al., 2022). In this paper, we provide theoretical guarantees on the convergence rate. In particular, we show that stochastic gradient descent (SGD) with teleportation converges to a basin of stationary points, where every point reachable by teleportation is also stationary. We also provide conditions under which one teleportation guarantees optimality of the entire gradient flow trajectory.

Previous applications of teleportation are limited to accelerating optimization. The second part of this paper explores a different objective – improving generalization. We relate properties of minima to their generalization ability and optimize them using teleportation. We empirically verify that certain sharpness metrics are correlated with generalization (Keskar et al., 2017), although teleporting towards flatter regions has negligible effects on the validation loss. Additionally, we hypothesize that generalization also depends on the curvature of minima. For fully connected networks, we derive an explicit expression for estimating curvatures and show that teleporting towards larger curvatures improves the model's generalizability.

To demonstrate the wide applicability of parameter space symmetry, we expand teleportation to standard optimization algorithms beyond SGD, including momentum, AdaGrad, RMSProp, and

Adam. Experimentally, teleportation improves the convergence speed for these algorithms. Inspired by conditional programming and optimization-based meta-learning (Andrychowicz et al., 2016), we also propose a meta-optimizer to learn where to move parameters in a loss level set. This approach avoids the computation cost of optimization on group manifolds and improves upon existing meta-learning methods that are restricted to local updates.

The convergence speedup, applications in improving generalization, and the ability to integrate with different optimizers demonstrate the potential of improving optimization using symmetry. In summary, our main contributions are:

- theoretical guarantees that teleportation accelerates the convergence rate of SGD;
- quantifying the curvature of a minimum and evidence of its correlation with generalization;
- a teleportation-based algorithm to improve generalization;
- various optimization algorithms with integrated teleportation including momentum, Ada-Grad, and optimization-based meta-learning.

## 2 RELATED WORK

**Parameter space symmetry.** Continuous symmetries have been identified in the parameter space of various architectures, including homogeneous activations (Badrinarayanan et al., 2015; Du et al., 2018), radial rescaling activations (Ganev et al., 2022), and softmax and batchnorm functions (Kunin et al., 2021). Permutation symmetry has been linked to the structure of minima (Şimşek et al., 2021; Entezari et al., 2022). Quiver representation theory provides a more general framework for symmetries in neural networks with pointwise (Armenta & Jodoin, 2021) and rescaling activations (Ganev & Walters, 2022). A new class of nonlinear and data-dependent symmetries are identified in (Zhao et al., 2023). Since symmetry defines transformations of parameters within a level set of the loss function, these works are the basis of the teleportation method discussed in our paper.

Knowledge of parameter space symmetry motivates new optimization methods. One line of work seeks algorithms that are invariant to symmetry transformations (Neyshabur et al., 2015; Meng et al., 2019). Others search in the orbit for parameters that can be optimized faster (Armenta et al., 2023; Zhao et al., 2022). We build on the latter by providing theoretical analysis on the improvement of the convergence rate and by augmenting the teleportation objective to improve generalization.

**Initializations and restarts.** Teleportation before training changes the initialization of parameters, which is known to affect the training dynamics. For example, imbalance between layers at initialization affects the convergence of gradient flows in two-layer models (Tarmoun et al., 2021). Different initializations, among other sources of variance, also lead to different model performance after convergence (Dodge et al., 2020; Bouthillier et al., 2021; Ramasinghe et al., 2022). In addition to initialization, teleportation allows changes in landscape multiple times throughout the training.

Teleportation during training re-initializes the parameters to a point with the same loss. Its effect can resemble warm restart (Loshchilov & Hutter, 2017), which encourages parameters to move to more stable regions by periodically increasing the learning rate. Compared to restarts, teleportation leads to smaller temporary increase in loss and provides more control of where to move the parameters.

**Sharpness of minima and generalization.** The sharpness of minima has been linked to the generalization ability of models both empirically and theoretically (Hochreiter & Schmidhuber, 1997; Keskar et al., 2017; Petzka et al., 2021; Ding et al., 2022; Zhou et al., 2020), which motivates optimization methods that find flatter minima (Chaudhari et al., 2017; Foret et al., 2021; Kwon et al., 2021; Kim et al., 2022). We employ teleportation to search for flatter points along the loss level sets. The sharpness of a minimum is often defined using properties of the Hessian of the loss function, such as the number of small eigenvalues (Keskar et al., 2017; Chaudhari et al., 2017; Sagun et al., 2017) or the product of the top $k$ eigenvalues (Wu et al., 2017). Alternatively, sharpness can be characterized by the maximum loss within a neighborhood of a minimum (Keskar et al., 2017; Foret et al., 2021; Kim et al., 2022) or approximated by the growth in the loss curve averaged over random directions (Izmailov et al., 2018). The sharpness of minima does not always capture generalization (Dinh et al., 2017) (Andriushchenko et al., 2023). Some reparametrizations do not affect generalization but can lead to minima with different sharpness.

## 3 THEORETICAL GUARANTEES FOR IMPROVING OPTIMIZATION

In this section, we provide a theoretical analysis of teleportation. We show that with teleportation, SGD converges to a basin of stationary points. Building on its relation to Newton's method, teleportation leads to a mixture of linear and quadratic convergence. Lastly, in certain loss functions, one teleportation guarantees optimality of the entire gradient flow trajectory.

**Symmetry Teleportation.** We briefly review the symmetry teleportation algorithm (Zhao et al., 2022), which searches for steeper points in a loss level set to accelerate gradient descent. Consider the optimization problem

$$\boldsymbol{w}^* = \arg\min_{\boldsymbol{w}\in\mathbb{R}^d} \mathcal{L}(\boldsymbol{w}), \quad \mathcal{L}(\boldsymbol{w}) \stackrel{\text{def}}{=} \mathbb{E}_{\xi\sim\mathcal{D}}\left[\mathcal{L}(\boldsymbol{w},\xi)\right]$$

where $\mathcal{D}$ is the data distribution, $\xi$ is data sampled from $\mathcal{D}$, $\mathcal{L}$ the loss, $\boldsymbol{w}$ the parameters of the model, and $\mathbb{R}^d$ the parameter space. Let $G$ be a group acting on the parameter space, such that

$$\mathcal{L}(\boldsymbol{w}) = \mathcal{L}(g \cdot \boldsymbol{w}), \quad \forall g \in G, \ \forall \boldsymbol{w} \in \mathbb{R}^d.$$

Symmetry teleportation uses gradient ascent to find the group element $g$ that maximizes the magnitude of the gradient, and applies $g$ to the parameters while leaving the loss value unchanged:

$$\boldsymbol{w}' = g \cdot \boldsymbol{w}, \quad g = \underset{g\in G}{\arg\max}\|\nabla\mathcal{L}(g \cdot \boldsymbol{w})\|^2.$$

### 3.1 TELEPORTATION AND SGD

At each iteration $t \in \mathbb{N}^+$ in SGD, we choose a group element $g^t \in G$ and use teleportation before each gradient step as follows

$$\boldsymbol{w}^{t+1} = g^t \cdot \boldsymbol{w}^t - \eta\nabla\mathcal{L}(g^t \cdot \boldsymbol{w}^t, \xi^t). \tag{1}$$

Here $\eta$ is a learning rate, $\nabla\mathcal{L}(\boldsymbol{w}^t, \xi^t)$ is the gradient of $\mathcal{L}(\boldsymbol{w}^t, \xi^t)$ with respect to the parameters $\boldsymbol{w}$, and $\xi^t \sim \mathcal{D}$ is a mini-batch of data sampled i.i.d from the data distribution at each iteration.

By choosing the group element that maximizes the gradient norm, we show in the following theorem that the iterates in equation 1 converge to a basin of stationary points, where all points that can be reached via teleportation are also stationary points (visualized in Figure 1).

**Theorem 3.1.** *(Smooth non-convex) Let $\mathcal{L}(\boldsymbol{w}, \xi)$ be $\beta$–smooth and let*

$$\sigma^2 \stackrel{\text{def}}{=} \mathcal{L}(\boldsymbol{w}^*) - \mathbb{E}\left[\inf_{\boldsymbol{w}}\mathcal{L}(\boldsymbol{w}, \xi)\right].$$

*Consider the iterates $\boldsymbol{w}^t$ given by equation 1 where*

$$g^t \in \arg\max_{g\in G}\|\nabla\mathcal{L}(g \cdot \boldsymbol{w}^t)\|^2,$$

*which we assume exists.* [1] *If $\eta = \frac{1}{\beta\sqrt{T-1}}$ then*

$$\min_{t=0,\ldots,T-1}\mathbb{E}\left[\max_{g\in G}\|\nabla\mathcal{L}(g \cdot \boldsymbol{w}^t)\|^2\right]$$
$$\leq \frac{2\beta}{\sqrt{T-1}}\mathbb{E}\left[\mathcal{L}(\boldsymbol{w}^0) - \mathcal{L}(\boldsymbol{w}^*)\right] + \frac{\beta\sigma^2}{\sqrt{T-1}}, \quad (2)$$

*where the expectation is the total expectation with respect to the data $\xi^t$ for $t = 0,\ldots,T-1$.*

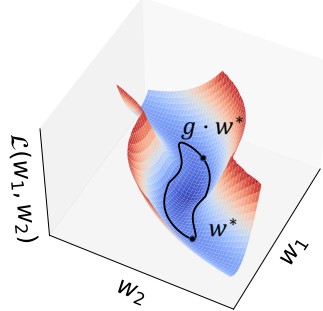

Figure 1: With teleportation, SGD converges to a basin where all points on the level set are stationary points.

This theorem is an improvement over vanilla SGD, for which we would have instead that

$$\min_{t=0,\ldots,T-1}\mathbb{E}\left[\|\nabla\mathcal{L}(\boldsymbol{w}^t)\|^2\right] \leq \frac{2\beta}{\sqrt{T-1}}\mathbb{E}\left[\mathcal{L}(\boldsymbol{w}^0) - \mathcal{L}(\boldsymbol{w}^*)\right] + \frac{\beta\sigma^2}{\sqrt{T-1}}.$$

---

[1]For instance when $G$ is compact and $\|\nabla\mathcal{L}(g \cdot \boldsymbol{w}^t)\|$ is continuous over $G$, or when the gradient is a coercive function and $G$ is bounded.

The above only guarantees that there exists a single point $\boldsymbol{w}^t$ for which the gradient norm will eventually be small. In contrast, our result in equation 2 guarantees that for all points over the orbit $\{g \cdot \boldsymbol{w}^t \ : \ \forall g \in G\}$, the gradient norm will be small. For strictly convex loss functions, $\max_{g \in G} \|\nabla \mathcal{L}(g \cdot \boldsymbol{w})\|^2$ is non-decreasing with $\mathcal{L}(\boldsymbol{w})$. In this case, the value of $\mathcal{L}$ is smaller after $T$ steps of SGD with teleportation, compared to vanilla SGD (Proposition A.2).

## 3.2 TELEPORTATION AND NEWTON'S METHOD

Intuitively, teleportation can speed up optimization as it behaves similarly to Newton's method. After a teleportation that takes parameters to a critical point on a level set, the gradient descent direction is the same as the Newton direction (Zhao et al., 2022). As a result, we can leverage the convergence of Newton's method to derive the convergence rate of teleportation for the deterministic setting.

**Proposition 3.2** (Quadratic term in convergence rate). *Let $\mathcal{L}$ be strictly convex and let $\boldsymbol{w}_0 \in \mathbb{R}^d$. Let*

$$\boldsymbol{w}' \in \underset{\boldsymbol{w} \in \mathbb{R}^d}{\arg \max} \frac{1}{2} \|\nabla \mathcal{L}(\boldsymbol{w})\|^2, \quad \text{s.t.} \quad \mathcal{L}(\boldsymbol{w}) = \mathcal{L}(\boldsymbol{w}_0).$$

*Let $\nabla^2 \mathcal{L}$ be the Hessian of $\mathcal{L}$, and $\lambda_{\max}(\nabla^2 \mathcal{L}(\boldsymbol{w}))$ be the largest eigenvalue of $\nabla^2 \mathcal{L}(\boldsymbol{w})$. If $\nabla \mathcal{L}(\boldsymbol{w}') \neq 0$, then there exists $\lambda_0$ such that $0 \leq \lambda_0 \leq \lambda_{\max}(\nabla^2 \mathcal{L}(\boldsymbol{w}_0))$, and one step of gradient descent after teleportation with learning rate $\gamma > 0$ gives*

$$\boldsymbol{w}_1 = \boldsymbol{w}' - \gamma \nabla \mathcal{L}(\boldsymbol{w}') = \boldsymbol{w}' - \gamma \lambda_0 \nabla^2 \mathcal{L}(\boldsymbol{w}')^{-1} \nabla \mathcal{L}(\boldsymbol{w}'). \tag{3}$$

*Let $\boldsymbol{w}' = g_0 \cdot \boldsymbol{w}_0$. If $\gamma \leq \frac{1}{\lambda_0}$, $\mathcal{L}$ is a $\mu$–strongly convex $L$–smooth function, and the Hessian is $G$–Lipschitz, then we have that*

$$\|\boldsymbol{w}_1 - \boldsymbol{w}^*\| \leq \frac{G}{2\mu} \|g_0 \cdot \boldsymbol{w}_0 - \boldsymbol{w}^*\|^2 + |1 - \gamma \lambda_0| \frac{L}{2\mu} \|g_0 \cdot \boldsymbol{w}_0 - \boldsymbol{w}^*\|.$$

More details about the assumptions and the proof are in Appendix B. Note that due to unknown step size $\lambda_0$, extra care is needed in establishing this convergence rate.

The above proposition shows that taking one step of teleportation and one gradient step, the result is equal to taking a dampened Newton step (equation 3). Hence, the convergence rate has a quadratically contracting term $\|g_0 \cdot \boldsymbol{w}_0 - \boldsymbol{w}^*\|^2$, which is typical of second order methods. In particular, setting $\gamma = 1/\lambda_0$ we would have local quadratic convergence. In contrast, without the teleportation step and under the same assumptions, we would have the following linear convergence

$$\|\boldsymbol{w}_1 - \boldsymbol{w}^*\| \leq (1 - \mu\gamma) \|\boldsymbol{w}_0 - \boldsymbol{w}^*\|$$

for $\gamma \leq \frac{1}{L}$ using gradient descent. Thus there would be no quadratically contracting term.

## 3.3 WHEN IS ONE TELEPORTATION ENOUGH

Despite the guaranteed improvement in convergence, teleporting before every gradient descent step is computationally expensive. Hence we teleport only occasionally. In fact, for certain optimization objectives, every point on the gradient flow has the largest gradient norm in its loss level set after one teleportation (Zhao et al., 2022). In past work, this result is limited to convex quadratic functions. In this section, we give a sufficient condition for when one teleportation results in an optimal trajectory for general loss functions. Full proofs can be found in Appendix C.

Let $V : \mathcal{M} \to T\mathcal{M}$ be a vector field on the manifold $\mathcal{M}$, where $T\mathcal{M}$ denotes the associated tangent bundle. Here we consider the parameter space $\mathcal{M} = \mathbb{R}^n$, although results in this section can be extended to optimization on other manifolds. In this case, we may write $V = v^i \frac{\partial}{\partial w^i}$ using the component functions $v^i : \mathbb{R}^n \to \mathbb{R}$ and coordinates $w^i$.

Consider a smooth loss function $\mathcal{L} : \mathcal{M} \to \mathbb{R}$. Let $G$ be a symmetry group of $\mathcal{L}$, i.e. $\mathcal{L}(g \cdot \boldsymbol{w}) = \mathcal{L}(\boldsymbol{w})$ for all $\boldsymbol{w} \in \mathcal{M}$ and $g \in G$. Let $\mathfrak{X}$ be the set of all vector fields on $\mathcal{M}$. Let $R = r^i \frac{\partial}{\partial w^i}$, where $r^i = -\frac{\partial \mathcal{L}}{\partial w_i}$, be the reverse gradient vector field. Let $\mathfrak{X}_\perp = \{A = a^i \frac{\partial}{\partial w^i} \in \mathfrak{X} | a^i \in C^\infty(\mathcal{M})$ and $\sum_i a^i(\boldsymbol{w}) r^i(\boldsymbol{w}) = 0, \forall \boldsymbol{w} \in \mathcal{M}\}$ be the set of vector fields orthogonal to $R$. If $G$ is a Lie group, the infinitesimal action of its Lie algebra $\mathfrak{g}$ defines a set of vector fields $\mathfrak{X}_\mathfrak{g} \subseteq \mathfrak{X}_\perp$.

A gradient flow is a curve $\gamma : \mathbb{R} \to \mathcal{M}$ where the velocity is given by the value of $R$, i.e. $\gamma'(t) = R_{\gamma(t)}$ for all $t \in \mathbb{R}$. The Lie bracket $[A, R]$ defines the derivative of $R$ with respect to $A$. Flows of $A$ and $R$ commute if and only if $[A, R] = 0$ (Theorem 9.44, Lee (2013)). That is, teleportation can affect the convergence rate only if $[A, R]\mathcal{L} \neq 0$ for some $A \in \mathfrak{X}_{\mathfrak{g}}$. To simplify notation, we write $([W, R]\mathcal{L})(\boldsymbol{w}) = 0$ for a set of vector fields $W \subseteq \mathfrak{X}$ when $([A, R]\mathcal{L})(\boldsymbol{w}) = 0$ for all $A \in W$.

We consider a gradient flow optimal if every point on the flow is a critical point of the magnitude of gradient in its loss level set. Note that this definition does not exclude the case where points on the flow are minimizers of the magnitude of gradient.

**Definition 3.3.** *Let $f : \mathcal{M} \to \mathbb{R}$, $\boldsymbol{w} \mapsto \left\| \frac{\partial \mathcal{L}}{\partial \boldsymbol{w}} \right\|_2^2$. A point $\boldsymbol{w} \in M$ is optimal with respect to a set of vector fields $W \subseteq \mathfrak{X}_\perp$ if $Af(\boldsymbol{w}) = 0$ for all $A \in W$. A gradient flow $\gamma : \mathbb{R} \to \mathcal{M}$ is optimal with respect to $W$ if $\gamma(t)$ is optimal with respect to $W$ for all $t \in \mathbb{R}$.*

**Proposition 3.4.** *A point $\boldsymbol{w} \in \mathcal{M}$ is optimal with respect to a set of vector fields $W$ if and only if $([W, R]\mathcal{L})(\boldsymbol{w}) = 0$.*

A sufficient condition for one teleportation to result in an optimal trajectory is that whenever the function $[A, R]\mathcal{L}$ vanishes at $\boldsymbol{w} \in \mathcal{M}$, it vanishes along the entire gradient flow starting at $\boldsymbol{w}$.

**Proposition 3.5.** *Let $W \subseteq \mathfrak{X}_\perp$ be a set of vector fields that are orthogonal to $\frac{\partial \mathcal{L}}{\partial \boldsymbol{w}}$. Assume that for all $\boldsymbol{w} \in \mathcal{M}$ such that $([W, R]\mathcal{L})(\boldsymbol{w}) = 0$, we have that $(R[W, R]\mathcal{L})(\boldsymbol{w}) = 0$. Then the gradient flow starting at any optimal point with respect to $W$ is optimal with respect to $W$.*

To help check when the assumption in Proposition 3.5 is satisfied, we provide an alternative form of $R[W, R]\mathcal{L}(\boldsymbol{w})$ when $[W, R]\mathcal{L}(\boldsymbol{w}) = 0$.

**Proposition 3.6.** *If at all optimal points in $S = \{ (M \frac{\partial \mathcal{L}}{\partial \boldsymbol{w}})^i \frac{\partial}{\partial w^i} \in \mathfrak{X} | \ M \in \mathbb{R}^{n \times n}, M^T = -M \}$,*

$$M_\alpha^j \frac{\partial \mathcal{L}}{\partial w_k} \frac{\partial \mathcal{L}}{\partial w_\alpha} \frac{\partial^3 \mathcal{L}}{\partial w^k \partial w_i \partial w^j} \frac{\partial \mathcal{L}}{\partial w^i} = 0$$

*for all anti-symmetric matrices $M \in \mathbb{R}^{n \times n}$, then the gradient flow starting at an optimal point in $S$ is optimal in $S$.*

From Proposition 3.6, we see that $R[W, R]\mathcal{L}(\boldsymbol{w})$ is not automatically 0 when $[W, R]\mathcal{L}(\boldsymbol{w}) = 0$. Therefore, even if the group is big enough to have its infinitesimal actions cover the tangent space of the level set ($\mathfrak{X}_{\mathfrak{g}} = \mathfrak{X}_\perp$), one teleportation does not guarantee that the gradient flow intersects all future level sets at optimal points. However, for loss functions that satisfy the condition in Proposition 3.5, teleporting once optimizes the entire trajectory. This is the case, for example, when $\frac{\partial^3 \mathcal{L}}{\partial w^k \partial w^i \partial w^j} \frac{\partial \mathcal{L}}{\partial w^\alpha} = \frac{\partial^3 \mathcal{L}}{\partial w^k \partial w^i \partial w^\alpha} \frac{\partial \mathcal{L}}{\partial w^j}$ for all $i, k, j, \alpha$ (Proposition C.3). In particular, all quadratic functions meet this condition.

## 4 TELEPORTATION FOR IMPROVING GENERALIZATION

Teleportation was originally proposed to speedup optimization. In this section, we explore the suitability of teleportation for improving generalization, which is another important aspect of deep learning. We first review definitions of the sharpness of minima. Then, we introduce a novel notion of the curvature of minima and discuss its implications on generalization. By observing how sharpness and curvature of minima are correlated with generalization, we improve generalization by incorporating sharpness and curvature into the objective for teleportation.

### 4.1 SHARPNESS OF MINIMA

Flat minima tend to generalize well (Hochreiter & Schmidhuber, 1997), typically characterized by numerous small Hessian eigenvalues. Although Hessian-based sharpness metrics are known to correlate well with generalization, they are expensive to compute and differentiate through. To use sharpness as an objective in teleportation, we consider changes in the loss averaged over random directions. Let $D$ be a set of vectors drawn randomly from the unit sphere $d_i \sim \{d \in \mathbb{R}^n : ||d|| = 1\}$, and $T$ a list of displacements $t_j \in \mathbb{R}$. Then, we have the following metric (Izmailov et al., 2018):

$$\text{Sharpness:} \quad \phi(\boldsymbol{w}, T, D) = \frac{1}{|T||D|} \sum_{t \in T} \sum_{d \in D} \mathcal{L}(\boldsymbol{w} + td). \tag{4}$$

## 4.2 CURVATURE OF MINIMA

At a minimum, the loss-invariant or flat directions are zero eigenvectors of the Hessian. The curvature along these directions does not directly affect Hessian-based sharpness metrics. However, these curvatures may affect generalization, by themselves or by correlating to the curvature along non-flat directions. Unlike the curvature of the loss (curve $\mathcal{L}(\boldsymbol{w})$ in Figure 2), the curvature of the minima (curve $\gamma$) is less well studied. We provide a novel method to quantify the curvature of the minima below.

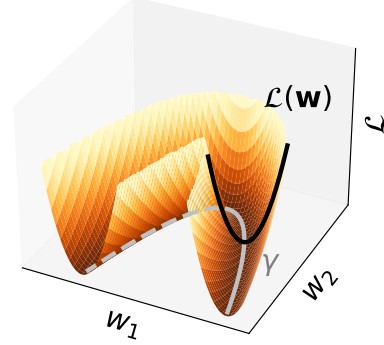

Assume that the loss function $\mathcal{L}$ has a $G$ symmetry. Consider the curve $\gamma_M : \mathbb{R} \times \mathbb{R}^n \to \mathbb{R}^n$ where $M \in \text{Lie}(G)$ and $\gamma_M(t, \boldsymbol{w}) = \exp(tM) \cdot \boldsymbol{w}$. Then $\gamma(0, \boldsymbol{w}) = \boldsymbol{w}$, and every point on $\gamma_M$ is in the minimum if $\boldsymbol{w}$ is a minimum. Let $\gamma' = \frac{d\gamma}{dt}$ be the derivative of a curve $\gamma$. The curvature of $\gamma$ is $\kappa(\gamma, t) = \frac{\|T'(t)\|}{\|\gamma'(t)\|}$, where $T(t) = \frac{\gamma'(t)}{\|\gamma'(t)\|}$ is the unit tangent vector. We assume that the action map is smooth, since calculating the curvature requires second

Figure 2: Gradient flow ($\mathcal{L}(\boldsymbol{w})$) and a curve on the minimum ($\gamma$). The curvature of both curves may affect generalization.

derivatives and optimizing the curvature via gradient descent requires third derivatives. For multilayer network with element-wise activations, we derive the group action, $\gamma$, and $\kappa$ in Appendix D.

Since the minimum can have more than one dimension, we measure the curvature of a point $\boldsymbol{w}$ on the minimum by averaging the curvature of $k$ curves with randomly selected Lie algebra elements $M_i \in \text{Lie}(G)$. The resulting new metric is

$$\text{Curvature:} \quad \psi(\boldsymbol{w}, k) = \frac{1}{k} \sum_{i=1}^{k} \kappa(\gamma_{M_i}(0, \boldsymbol{w}), 0) . \tag{5}$$

There are different ways to measure the curvature of a higher-dimensional manifold, such as using the Gaussian curvature of 2D subspaces of the tangent space. However, our method of approximating the mean curvature is easier to compute and suitable as a differentiable objective.

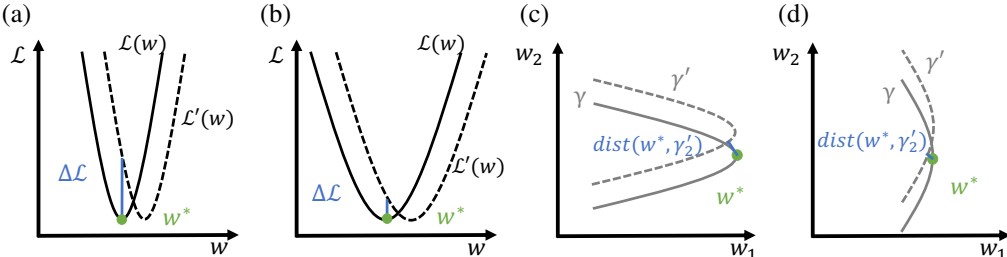

Figure 3: Illustration of the effect of sharpness (a,b) and curvature (c,d) of minima on generalization. See Figure 2 for a 3D visualization of the curves $\mathcal{L}(\boldsymbol{w})$ and $\gamma$. When the loss landscape shifts due to a change in data distribution, sharper minima have larger increase in loss. In the example shown, minima with larger curvature moves further away from the shifted minima.

## 4.3 CORRELATION WITH GENERALIZATION

Generalization reflects how loss changes with shifts in data distribution. The sharpness of minima is well known to be correlated with generalization. Figure 3(a)(b) visualizes an example of the shift in loss landscape ($\mathcal{L}(\boldsymbol{w})$), and the change of loss $\Delta\mathcal{L}$ at a minimizer $\boldsymbol{w}^*$ is large when the minimum is sharp. The relation between the curvature of minimum and generalization is less well studied. Figure 3(c)(d) shows one possible shift of the minimum ($\gamma$). Under this shifting, the minimizer with a larger curvature becomes farther away from the shifted minimum. The curve on the minimum can shift in other directions. Appendix E.2 provides analytical examples of the correlation between curvature and expected distance between the old and shifted minimum.

Table 1: Correlation with validation loss

| | sharpness ($\phi$) | | | curvature ($\psi$) | |
|---|---|---|---|---|---|
| MNIST | Fashion-MNIST | CIFAR-10 | MNIST | Fashion-MNIST | CIFAR-10 |
| 0.704 | 0.790 | 0.899 | -0.050 | -0.232 | -0.167 |

We verify the correlation between sharpness, curvatures, and validation loss on MNIST (Deng, 2012), Fashion-MNIST (Xiao et al., 2017), and CIFAR-10 (Krizhevsky et al., 2009). On each dataset, we train 100 three-layer neural networks with LeakyReLU using different initializations. Details of the setup can be found in Appendix E.3.

Table 1 shows the Pearson correlation between validation loss and sharpness or curvature (scatter plots in Figure 9 and 10 in the appendix). In all three datasets, sharpness has a strong positive correlation with validation loss, meaning that the average change in loss under perturbations is a good indicator of test performance. For the architecture we consider, the curvature of minima is negatively correlated with the validation loss. We observe that the magnitudes of the curvatures are small, which suggests that the minima are relatively flat.

### 4.4 TELEPORTATION FOR IMPROVING GENERALIZATION

To improve the generalization ability of the minimizer and to gain understanding of the curvature of minima, we teleport parameters to regions with different sharpness and curvature. Multi-layer neural networks have $GL(\mathbb{R})$ symmetry between layers (Appendix D.1). We parametrize the group by its Lie algebra $T$, and perform gradient ascent on $T$ to maximize the gradient norm at the transformed parameters $|\nabla L|_{\exp(T) \cdot w}|$. Algorithm 2 in Appendix E.4 demonstrates how to increase curvature $\psi$ by teleporting two layers, with hidden dimension $h$, in an MLP. In experiments, we use an extended version of the algorithm, which teleports all layers by optimizing on a list of $T$'s concurrently. During teleportation, we perform gradient descent on the group elements to change $\phi$ or $\psi$. Results are averaged over 5 runs.

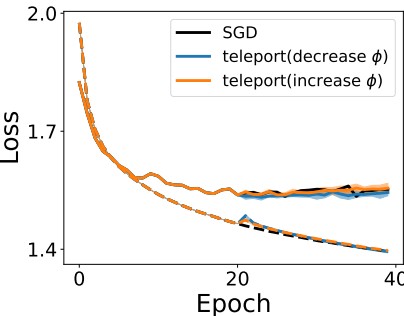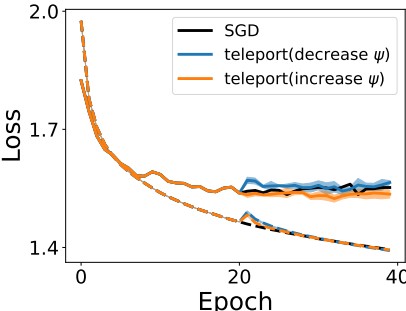

Figure 4: Changing sharpness (left) or curvature (right) using teleportation and its effect on generalization on CIFAR-10. Solid line represents average test loss, and dashed line represent average training loss. Teleporting to decrease sharpness improves validation loss slightly. Teleportation changing curvatures has a more significant impact on generalization ability.

Figure 4 shows the training curve of SGD on CIFAR-10, with one teleportation at epoch 20. Similar results for AdaGrad can be found in Appendix E.4. Teleporting to flatter points slightly improves the validation loss, while teleporting to sharper points has no effect. Since the group action keeps the loss invariant only on the batch of data used in teleportation, the errors incurred in teleportation have a similar effect to a warm restart, which makes the effect of changing sharpness less clear.

Interestingly, by changing the curvature, teleportation is able to affect generalization. Teleporting to points with larger curvatures helps find a minimum with lower validation loss, while teleporting to points with smaller curvatures has the opposite effect. This suggests that at least locally, curvature is correlated with generalization. Details of the experiment setup can be found in Appendix E.4.

## 5 APPLICATIONS TO OTHER OPTIMIZATION ALGORITHMS

Having shown teleportation's potential to improve optimization and generalization, we demonstrate its wide applicability by integrating teleportation into different optimizers and meta-learning.

### 5.1 STANDARD OPTIMIZERS

Teleportation improves optimization not only for SGD. To show that teleportation works well with other standard optimizers, we train a 3-layer neural network on MNIST using different optimizers with and without teleportation. During training, we teleport once at the first epoch, using 8 minibatches of size 200. Details can be found in Appendix F.2.

Figure 5 shows that teleportation improves the convergence rate when using AdaGrad, SGD with momentum, RMSProp, and Adam. The runtime for a teleportation is smaller than the time required to train one epoch, hence teleportation improves convergence rate per epoch at almost no additional cost of time (Figure 13 in the appendix).

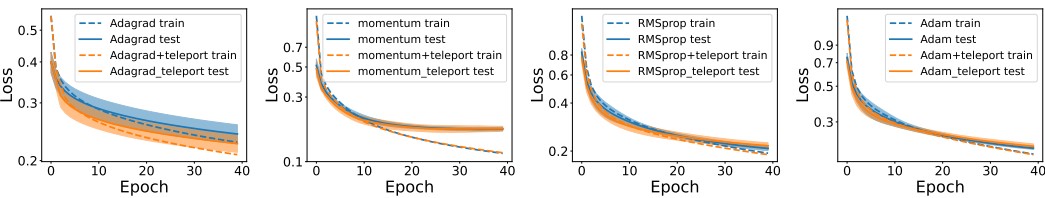

Figure 5: Integrating teleportation with AdaGrad, momentum, RMSProp, and Adam improves the convergence rate on MNIST. Solid line represents the average test loss, and dashed line represents the average training loss. Shaded areas are 1 standard deviation of the test loss across 5 runs.

### 5.2 LEARNING TO TELEPORT

In optimization-based meta-learning, the parameter update rule or the hyperparameters are learned using a meta-optimizer (Andrychowicz et al., 2016; Finn et al., 2017). Teleportation introduces an additional degree of freedom in parameter updates. We augment existing meta-learning algorithms by learning both the local update and teleportation. This allows us to teleport without implementing the additional optimization step on groups, which reduces computation time.

Let $\boldsymbol{w}_t \in \mathbb{R}^d$ be the parameters at time $t$, and $\nabla_t = \frac{\partial \mathcal{L}}{\partial \boldsymbol{w}}\big|_{\boldsymbol{w}_t}$ be the gradient of the loss $\mathcal{L}$. In gradient descent, the update rule with learning rate $\eta$ is

$$\boldsymbol{w}_{t+1} = \boldsymbol{w}_t - \eta \nabla_t.$$

In meta-learning (Andrychowicz et al., 2016), the update on $\boldsymbol{w}_t$ is learned using a meta-learning optimizer $m$, which takes $\nabla_t$ as input. Here $m$ is an LSTM model. Denote $h_t$ as the hidden state in the LSTM and $\phi$ as the parameters in $m$. The update rule is

$$\boldsymbol{w}_{t+1} = \boldsymbol{w}_t + f_t$$
$$\begin{bmatrix} f_t \\ h_{t+1} \end{bmatrix} = m(\nabla_t, h_t, \phi).$$

Extending this approach beyond an additive update rule, we learn to teleport. Let $G$ be a group whose action on the parameter space leaves $\mathcal{L}$ invariant. We use two meta-learning optimizers $m_1, m_2$ to learn the update direction $f_t \in \mathbb{R}^d$ and the group element $g_t \in G$:

$$\boldsymbol{w}_{t+1} = g_t \cdot (\boldsymbol{w}_t + f_t)$$
$$\begin{bmatrix} f_t \\ h_{1_{t+1}} \end{bmatrix} = m_1(\nabla_t, h_{1_t}, \phi_1), \quad \begin{bmatrix} g_t \\ h_{2_{t+1}} \end{bmatrix} = m_2(\nabla_t, h_{2_t}, \phi_2).$$

**Experiment setup.** We train and test on two-layer neural networks $\mathcal{L}(W_1, W_2) = \|Y - W_2\sigma(W_1X)\|_2$, where $W_2, W_1, X, Y \in \mathbb{R}^{20 \times 20}$, and $\sigma$ is the LeakyReLU function with slope coefficient 0.1. Both meta-optimizers are two-layer LSTMs with hidden dimension 300. We train the meta-optimizers on multiple trajectories created with different initializations, each consisting of 100 steps of gradient descent on $\mathcal{L}$ with random $X, Y$ and randomly initialized $W$'s. We update the parameters in $m_1$ and $m_2$ by unrolling every 10 steps. The learning rate for meta-optimizers are $10^{-4}$ for $m_1$ and $10^{-3}$ for $m_2$. We test the meta-optimizers using 5 trajectories not seen in training.

Algorithm 1 summarizes the training procedure. The vanilla gradient descent baseline ("GD") uses the largest learning rate that does not lead to divergence ($3 \times 10^{-4}$). The second baseline ("LSTM(update)") learns the update $f_t$ only and does not perform teleportation ($g_t = I, \forall t$). The third baseline ("LSTM(lr,tele)") learns the group element $g_t$ and the learning rate used to perform gradient descent instead of the update $f_t$. We keep training until adding more training trajectories does not improve convergence rate. We use 700 training trajectories for our approach, 600 for the second baseline, and 30 for the third baseline.

**Results.** By learning both the local update $f_t$ and non-local transformation $g_t$, our meta-optimizer successfully learns to learn faster. Figure 6 shows the improvement of our approach from the previous meta-learning method, which only learns $f_t$. Compared to the baselines, learning the two types of updates together ("LSTM(update,tele)") achieves better convergence rate than learning them separately. Additionally, learning the group element $g_t$ eliminates the need for performing gradient ascent on the group manifold and reduces hyperparameter tuning for teleportation. As an example of successful integration of teleportation into existing optimization algorithms, this toy experiment demonstrates the flexibility and promising applications of teleportation.

---

**Algorithm 1** Learning to teleport

---

**Input:** Loss function $\mathcal{L}$, learning rate $\eta$, number of epochs $T$, LSTM models $m_1, m_2$ with initial parameters $\phi_1, \phi_2$, unroll step $t_{unroll}$.
**Output:** Trained parameters $\phi_1$ and $\phi_2$.
**for** each training initialization **do**
    **for** $t = 1$ **to** $T$ **do**
        $f_t, h_{1_{t+1}} = m_1(\nabla_t, h_{1_t}, \phi_1)$
        $g_t, h_{2_{t+1}} = m_2(\nabla_t, h_{2_t}, \phi_2)$
        $\boldsymbol{w} \leftarrow g_t \cdot (\boldsymbol{w} + f_t)$
        **if** $t \mod t_{unroll} = 0$ **then**
            update $\phi_1, \phi_2$ by back-propogation from the accumulated loss $\sum_{i=t-t_{unroll}}^{t} \mathcal{L}(\boldsymbol{w}_i)$
        **end if**
    **end for**
**end for**

---

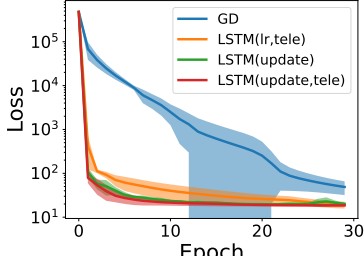

Figure 6: Performance of the trained meta-optimizer on the test set. Learning both local update $f_t$ and nonlocal transformation $g_t$ results in better convergence rate than learning only local updates or learning only teleportation.

# 6 DISCUSSION

Teleportation is a powerful tool to search in the loss level sets for parameters with desired properties. We provide theoretical guarantees that teleportation accelerates the convergence rate of SGD. Using concepts in symmetry, we propose a novel notion of curvature and show that incorporating additional teleportation objectives such as changing the curvatures can be beneficial to generalization. The close relationship between symmetry and optimization opens up a number of exciting opportunities. Exploring other objectives in teleportation appears to be an interesting future direction. Other possible applications include extending teleportation to different architectures, such as convolutional or graph neural networks, and to different algorithms, such as sampling-based optimization.

The empirical results linking sharpness and curvatures to generalization are intriguing. However, the theoretical origin of their relation remains unclear. In particular, a precise description of how the loss landscape changes under distribution shifts is not known. More investigation of the correlation between curvatures and generalization will help teleportation to further improve generalization and take us a step closer to understanding the loss landscape.

ACKNOWLEDGEMENTS

This work was supported in part by Army-ECASE award W911NF-23-1-0231, the U.S. Department Of Energy, Office of Science under #DE-SC0022255, IARPA HAYSTAC Program, CDC-RFA-FT-23-0069, NSF Grants #2205093, #2146343, #2134274, #2107256, and #2134178.

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

## APPENDIX

This appendix contains proofs, experiment setups, as well as additional results and discussions. Appendix A through C contain proofs for theoretical results in Section 3. Appendix D provides details about curves induced by symmetry and the curvature of the minimum. Appendix E discusses possible theoretical approaches to relate curvatures and generalization. This section also contains experiment details on computing correlations and the algorithm that uses teleportation to change curvature. Appendix F describes experiment setups and different strategies of integrating teleportation into various optimization algorithms.

The code used for our experiments is available at: `https://github.com/Rose-STL-Lab/Teleportation-Optimization`.

## A  TELEPORTATION AND SGD

This section includes a proof for Theorem 3.1. Additionally, we discuss the theorem's implication when the loss function is strictly convex.

**Lemma A.1** (Descent Lemma). *Let $\mathcal{L}(\boldsymbol{w}, \xi)$ be a $\beta$–smooth function. It follows that*

$$\mathbb{E}\left[\|\nabla\mathcal{L}(\boldsymbol{w},\xi)\|^2\right] \leq 2\beta(\mathcal{L}(\boldsymbol{w}) - \mathcal{L}(\boldsymbol{w}^*)) + 2\beta(\mathcal{L}(\boldsymbol{w}^*) - \mathbb{E}\left[\inf_{\boldsymbol{w}}\mathcal{L}(\boldsymbol{w},\xi)\right]). \tag{6}$$

*Proof.* Since $\mathcal{L}(w, \xi)$ is smooth we have that

$$\mathcal{L}(\boldsymbol{z},\xi) - \mathcal{L}(\boldsymbol{w},\xi) \leq \langle\nabla\mathcal{L}(\boldsymbol{w},\xi), \boldsymbol{z} - \boldsymbol{w}\rangle + \frac{\beta}{2}\|\boldsymbol{z} - \boldsymbol{w}\|^2, \quad \forall \boldsymbol{z}, \boldsymbol{w} \in \mathbb{R}^d. \tag{7}$$

By inserting

$$\boldsymbol{z} = \boldsymbol{w} - \frac{1}{\beta}\nabla\mathcal{L}(\boldsymbol{w},\xi)$$

into equation 7 we have that

$$\mathcal{L}\big(\boldsymbol{w} - (1/\beta)\nabla\mathcal{L}(\boldsymbol{w},\xi),\xi\big) \leq \mathcal{L}(\boldsymbol{w},\xi) - \frac{1}{2\beta}\|\nabla\mathcal{L}(\boldsymbol{w},\xi)\|^2. \tag{8}$$

Re-arranging we have that

$$\begin{aligned}
\mathcal{L}(\boldsymbol{w}^*,\xi) - \mathcal{L}(\boldsymbol{w},\xi) &= \mathcal{L}(\boldsymbol{w}^*,\xi) - \inf_{\boldsymbol{w}}\mathcal{L}(\boldsymbol{w},\xi) + \inf_{\boldsymbol{w}}\mathcal{L}(\boldsymbol{w},\xi) - \mathcal{L}(\boldsymbol{w},\xi) \\
&\leq \mathcal{L}(\boldsymbol{w}^*,\xi) - \inf_{\boldsymbol{w}}\mathcal{L}(\boldsymbol{w},\xi) + \mathcal{L}\big(\boldsymbol{w} - (1/\beta)\nabla\mathcal{L}(\boldsymbol{w},\xi),\xi\big) - \mathcal{L}(\boldsymbol{w},\xi) \\
&\overset{equation\ 8}{\leq} \mathcal{L}(\boldsymbol{w}^*,\xi) - \inf_{\boldsymbol{w}}\mathcal{L}(\boldsymbol{w},\xi) - \frac{1}{2\beta}\|\nabla\mathcal{L}(\boldsymbol{w},\xi)\|^2,
\end{aligned}$$

where the first inequality follows because $\inf_{\boldsymbol{w}}\mathcal{L}(\boldsymbol{w},\xi) \leq \mathcal{L}(\boldsymbol{w},\xi), \forall\boldsymbol{w}$. Re-arranging the above and taking expectation gives

$$\begin{aligned}
\mathbb{E}\left[\|\nabla\mathcal{L}(\boldsymbol{w},\xi)\|^2\right] &\leq 2\mathbb{E}\left[\beta(\mathcal{L}(\boldsymbol{w}^*,\xi) - \inf_{\boldsymbol{w}}\mathcal{L}(\boldsymbol{w},\xi) + \mathcal{L}(\boldsymbol{w},\xi) - \mathcal{L}(\boldsymbol{w}^*,\xi))\right] \\
&\leq 2\beta\mathbb{E}\left[\mathcal{L}(\boldsymbol{w}^*,\xi) - \inf_{\boldsymbol{w}}\mathcal{L}(\boldsymbol{w},\xi) + \mathcal{L}(\boldsymbol{w},\xi) - \mathcal{L}(\boldsymbol{w}^*,\xi)\right] \\
&\leq 2\beta(\mathcal{L}(\boldsymbol{w}) - \mathcal{L}(\boldsymbol{w}^*)) + 2\beta(\mathcal{L}(\boldsymbol{w}^*) - \mathbb{E}\left[\inf_{\boldsymbol{w}}\mathcal{L}(\boldsymbol{w},\xi)\right]).
\end{aligned}$$

$\square$

At each iteration $t \in \mathbb{N}^+$ in SGD, we choose a group element $g^t \in G$ and use teleportation before each gradient step as follows

$$\boldsymbol{w}^{t+1} = g^t \cdot \boldsymbol{w}^t - \eta\nabla\mathcal{L}(g^t \cdot \boldsymbol{w}^t, \xi^t). \tag{9}$$

Here $\eta$ is a learning rate, $\nabla\mathcal{L}(\boldsymbol{w}^t, \xi^t)$ is a gradient of $\mathcal{L}(\boldsymbol{w}^t, \xi^t)$ with respect to the parameters $\boldsymbol{w}$, and $\xi^t \sim \mathcal{D}$ is a mini-batch of data sampled i.i.d at each iteration.

**Theorem 3.1.** *Let $\mathcal{L}(\boldsymbol{w}, \xi)$ be $\beta$–smooth and let*

$$\sigma^2 \overset{def}{=} \mathcal{L}(\boldsymbol{w}^*) - \mathbb{E}\left[\inf_{\boldsymbol{w}} \mathcal{L}(\boldsymbol{w}, \xi)\right].$$

*Consider the iterates $\boldsymbol{w}^t$ given by equation 1 where*

$$g^t \in \arg\max_{g \in G} \|\nabla \mathcal{L}(g \cdot \boldsymbol{w}^t)\|^2. \tag{10}$$

*If $\eta = \frac{1}{\beta\sqrt{T-1}}$ then*

$$\min_{t=0,\dots,T-1} \mathbb{E}\left[\max_{g \in G} \|\nabla \mathcal{L}(g \cdot \boldsymbol{w}^t)\|^2\right] \le \frac{2\beta}{\sqrt{T-1}} \mathbb{E}\left[\mathcal{L}(\boldsymbol{w}^0) - \mathcal{L}(\boldsymbol{w}^*)\right] + \frac{\beta\sigma^2}{\sqrt{T-1}}. \tag{11}$$

*Proof.* First note that if $\mathcal{L}(\boldsymbol{w}, \xi)$ is $\beta$–smooth, then $\mathcal{L}(\boldsymbol{w})$ is also a $\beta$–smooth function, that is

$$\mathcal{L}(\boldsymbol{z}) - \mathcal{L}(\boldsymbol{w}) - \langle \nabla \mathcal{L}(\boldsymbol{w}), \boldsymbol{z} - \boldsymbol{w} \rangle \le \frac{\beta}{2} \|\boldsymbol{z} - \boldsymbol{w}\|^2. \tag{12}$$

Using equation 1 with $\boldsymbol{z} = \boldsymbol{w}^{t+1}$ and $\boldsymbol{w} = g^t \cdot \boldsymbol{w}^t$, together with equation 12 and the fact that the group action preserves loss, we have that

$$\mathcal{L}(\boldsymbol{w}^{t+1}) \le \mathcal{L}(g^t \cdot \boldsymbol{w}^t) + \left\langle \nabla \mathcal{L}(g^t \cdot \boldsymbol{w}^t), \boldsymbol{w}^{t+1} - g^t \cdot \boldsymbol{w}^t \right\rangle + \frac{\beta}{2} \|\boldsymbol{w}^{t+1} - g^t \cdot \boldsymbol{w}^t\|^2 \tag{13}$$

$$= \mathcal{L}(\boldsymbol{w}^t) - \eta_t \left\langle \nabla \mathcal{L}(g^t \cdot \boldsymbol{w}^t), \nabla \mathcal{L}(g^t \cdot \boldsymbol{w}^t, \xi^t) \right\rangle + \frac{\beta\eta_t^2}{2} \|\nabla \mathcal{L}(g^t \cdot \boldsymbol{w}^t, \xi^t)\|^2. \tag{14}$$

Taking expectation conditioned on $\boldsymbol{w}^t$, we have that

$$\mathbb{E}_t\left[\mathcal{L}(\boldsymbol{w}^{t+1})\right] \le \mathcal{L}(\boldsymbol{w}^t) - \eta_t \|\nabla \mathcal{L}(g^t \cdot \boldsymbol{w}^t)\|^2 + \frac{\beta\eta_t^2}{2} \mathbb{E}_t\left[\|\nabla \mathcal{L}(g^t \cdot \boldsymbol{w}^t, \xi^t)\|^2\right]. \tag{15}$$

Now since $\mathcal{L}(\boldsymbol{w}, \xi)$ is $\beta$–smooth, from Lemma A.1 above we have that

$$\mathbb{E}\left[\|\nabla \mathcal{L}(\boldsymbol{w}, \xi)\|^2\right] \le 2\beta(\mathcal{L}(\boldsymbol{w}) - \mathcal{L}(\boldsymbol{w}^*)) + 2\beta(\mathcal{L}(\boldsymbol{w}^*) - \mathbb{E}\left[\inf_{\boldsymbol{w}} \mathcal{L}(\boldsymbol{w}, \xi)\right]) \tag{16}$$

Using equation 16 with $\boldsymbol{w} = g^t \circ \boldsymbol{w}^t$ we have that

$$\mathbb{E}_t\left[\mathcal{L}(\boldsymbol{w}^{t+1})\right] \le \mathcal{L}(\boldsymbol{w}^t) - \eta_t \|\nabla \mathcal{L}(g^t \cdot \boldsymbol{w}^t)\|^2$$
$$+ \beta^2 \eta_t^2 \left(\mathcal{L}(g^t \cdot \boldsymbol{w}^t) - \mathcal{L}(\boldsymbol{w}^*) + \mathcal{L}(\boldsymbol{w}^*) - \mathbb{E}\left[\inf_{\boldsymbol{w}} \mathcal{L}(\boldsymbol{w}, \xi)\right]\right). \tag{17}$$

Using that $\mathcal{L}(g^t \cdot \boldsymbol{w}^t) = \mathcal{L}(\boldsymbol{w}^t)$, taking full expectation and re-arranging terms gives

$$\eta_t \mathbb{E}\left[\|\nabla \mathcal{L}(g^t \cdot \boldsymbol{w}^t)\|^2\right] \le (1 + \beta^2 \eta_t^2) \mathbb{E}\left[\mathcal{L}(\boldsymbol{w}^t) - \mathcal{L}^*\right] - \mathbb{E}\left[\mathcal{L}(\boldsymbol{w}^{t+1}) - \mathcal{L}^*\right] + \beta^2 \eta_t^2 \sigma^2. \tag{18}$$

Now we use a re-weighting trick introduced in Stich (2019). Let $\alpha_t > 0$ be a sequence such that $\alpha_t(1 + \beta^2 \eta_t^2) = \alpha_{t-1}$. Consequently if $\alpha_{-1} = 1$ then $\alpha_t = (1 + \beta^2 \eta_t^2)^{-(t+1)}$. Multiplying by both sides of equation 18 by $\alpha_t$ thus gives

$$\alpha_t \eta_t \mathbb{E}\left[\|\nabla \mathcal{L}(g^t \cdot \boldsymbol{w}^t)\|^2\right] \le \alpha_{t-1} \mathbb{E}\left[\mathcal{L}(\boldsymbol{w}^t) - \mathcal{L}^*\right] - \alpha_t \mathbb{E}\left[\mathcal{L}(\boldsymbol{w}^{t+1}) - \mathcal{L}^*\right] + \alpha_t \beta^2 \eta_t^2 \sigma^2. \tag{19}$$

Summing up from $t = 0, \dots, T-1$, and using telescopic cancellation, gives

$$\sum_{t=0}^{T-1} \alpha_t \eta_t \mathbb{E}\left[\|\nabla \mathcal{L}(g^t \cdot \boldsymbol{w}^t)\|^2\right] \le \mathbb{E}\left[\mathcal{L}(\boldsymbol{w}^0) - \mathcal{L}^*\right] + \beta^2 \sigma^2 \sum_{t=0}^{T-1} \alpha_t \eta_t^2 \tag{20}$$

Let $A = \sum_{t=0}^{T-1} \alpha_t \eta_t$. Dividing both sides by $A$ gives

$$\min_{t=0,\dots,T-1} \mathbb{E}\left[\|\nabla \mathcal{L}(g^t \cdot \boldsymbol{w}^t)\|^2\right] \le \frac{1}{\sum_{t=0}^{T-1} \alpha_t \eta_t} \sum_{t=0}^{T-1} \alpha_t \eta_t \|\nabla \mathcal{L}(g^t \cdot \boldsymbol{w}^t)\|^2$$

$$\le \frac{\mathbb{E}\left[\mathcal{L}(\boldsymbol{w}^0) - \mathcal{L}^*\right] + \beta^2 \sigma^2 \sum_{t=0}^{T-1} \alpha_t \eta_t^2}{\sum_{t=0}^{T-1} \alpha_t \eta_t}. \tag{21}$$

Finally, if $\eta_t \equiv \eta$ then

$$\sum_{t=0}^{T-1} \alpha_t \eta_t = \eta \sum_{t=0}^{T-1} (1 + \beta^2 \eta_t^2)^{-(t+1)} = \frac{\eta}{1 + \beta^2 \eta^2} \frac{1 - (1 + \beta^2 \eta^2)^{-T}}{1 - (1 + \beta^2 \eta^2)^{-1}} \tag{22}$$

$$= \frac{1 - (1 + \beta^2 \eta^2)^{-T}}{\beta^2 \eta} \tag{23}$$

To bound the term with the $-T$ power, we use that

$$(1 + \beta^2 \eta^2)^{-T} \leq \frac{1}{2} \quad \Longrightarrow \quad \frac{\log(2)}{\log(1 + \beta^2 \eta^2)} \leq T.$$

To simplify the above expression we can use

$$\frac{x}{1 + x} \leq \log(1 + x) \leq x, \quad \text{for } x \geq -1,$$

thus

$$\frac{\log(2)}{\log(1 + \beta^2 \eta^2)} \leq \frac{1 + \beta^2 \eta^2}{\beta^2 \eta^2} \leq T.$$

Using the above we have that

$$\sum_{t=0}^{T-1} \alpha_t \eta_t \geq \frac{1}{2\beta^2 \eta}, \quad \text{for } T \geq \frac{1 + \beta^2 \eta^2}{\beta^2 \eta^2}$$

Using this lower bound in equation 21 gives

$$\min_{t=0,\ldots,T-1} \mathbb{E}\left[\|\nabla \mathcal{L}(g^t \cdot \boldsymbol{w}^t)\|^2\right] \leq 2\beta^2 \eta \mathbb{E}\left[\mathcal{L}(\boldsymbol{w}^0) - \mathcal{L}^*\right] + \eta \beta^2 \sigma^2, \quad \text{for } T \geq \frac{1 + \beta^2 \eta^2}{\beta^2 \eta^2}.$$

Now note that

$$T \geq \frac{1 + \beta^2 \eta^2}{\beta^2 \eta^2} \Leftrightarrow \beta^2 \eta^2 (T - 1) \geq 1 \Leftrightarrow \eta \geq \frac{1}{\beta \sqrt{(T-1)}}.$$

Thus finally setting $\eta = \frac{1}{\beta\sqrt{T-1}}$ gives the result equation 2.

$\square$

**Proposition A.2.** *Assume that $\mathcal{L} : \mathbb{R}^n \to \mathbb{R}$ is strictly convex and twice continuously differentiable. Assume also that for any two points $\boldsymbol{w}_a, \boldsymbol{w}_b \in \mathbb{R}^n$ such that $\mathcal{L}(\boldsymbol{w}_a) = \mathcal{L}(\boldsymbol{w}_b)$, there exists a $g \in G$ such that $\boldsymbol{w}_a = g \cdot \boldsymbol{w}_b$. At two points $\boldsymbol{w}_1, \boldsymbol{w}_2 \in \mathbb{R}^n$, if $\max_{g \in G} \|\nabla \mathcal{L}(g \cdot \boldsymbol{w}_1)\|^2 = \|\nabla \mathcal{L}(\boldsymbol{w}_2)\|^2$, then $\mathcal{L}(\boldsymbol{w}_1) \leq \mathcal{L}(\boldsymbol{w}_2)$.*

*Proof.* Let $S(x) = \{\boldsymbol{w} : \mathcal{L}(\boldsymbol{w}) = x\}$ be the level sets of $\mathcal{L}$, and $X = \{\mathcal{L}(\boldsymbol{w}) : \boldsymbol{w} \in \mathbb{R}^n\}$ be the image of $\mathcal{L}$. Since $G$ acts transitively on the level sets of $\mathcal{L}$, $\max_{g \in G} \|\nabla \mathcal{L}(g \cdot \boldsymbol{w})\|^2 = \max_{\boldsymbol{w} \in S(x)} \|\nabla \mathcal{L}(\boldsymbol{w})\|^2$. To simplify notation, we define a function $F : X \to \mathbb{R}$, $F(x) = \max_{\boldsymbol{w} \in S(x)} \|\nabla \mathcal{L}(\boldsymbol{w})\|^2$. Since $\nabla \mathcal{L}(\boldsymbol{w})$ is continuously differentiable, the directional derivative of $F$ is defined. Additionally, since $\mathcal{L}$ is continuous and its domain $\mathbb{R}^n$ is connected, its image $X$ is also connected. This means that for any $\boldsymbol{w}_1, \boldsymbol{w}_2 \in \mathbb{R}^n$ and $\min(\mathcal{L}(\boldsymbol{w}_1), \mathcal{L}(\boldsymbol{w}_2)) \leq y \leq \max(\mathcal{L}(\boldsymbol{w}_1), \mathcal{L}(\boldsymbol{w}_2))$, there exists a $\boldsymbol{w}_3 \in \mathbb{R}^n$ such that $\mathcal{L}(\boldsymbol{w}_3) = y$.

Next, we show that $F(\cdot)$ is strictly increasing by contradiction.

Suppose that $\mathcal{L}(\boldsymbol{w}_1) < \mathcal{L}(\boldsymbol{w}_2)$ and $F(\mathcal{L}(\boldsymbol{w}_1)) \geq F(\mathcal{L}(\boldsymbol{w}_2))$. By the mean value theorem, there exists a $\boldsymbol{w}_3$ such that $\mathcal{L}(\boldsymbol{w}_1) < \mathcal{L}(\boldsymbol{w}_3) < \mathcal{L}(\boldsymbol{w}_2)$ and the directional derivative of $F$ in the direction towards $\mathcal{L}(\boldsymbol{w}_2)$ is non-positive: $\partial_{\mathcal{L}(\boldsymbol{w}_2) - \mathcal{L}(\boldsymbol{w}_3)} F(\mathcal{L}(\boldsymbol{w}_3)) \leq 0$. Let $\boldsymbol{w}_3^* \in \arg\max_{\boldsymbol{w} \in S(\mathcal{L}(\boldsymbol{w}_3))} \|\nabla \mathcal{L}(\boldsymbol{w})\|^2$ be a point that has the largest gradient norm in $S(\mathcal{L}(\boldsymbol{w}_3))$. Then at $\boldsymbol{w}_3^*$, $\|\nabla \mathcal{L}\|^2$ cannot increase along the gradient direction. However, this means

$$\nabla \mathcal{L}(\boldsymbol{w}_3^*) \cdot \frac{\partial}{\partial \boldsymbol{w}} \|\nabla \mathcal{L}(\boldsymbol{w}_3^*)\|^2 = \nabla \mathcal{L}(\boldsymbol{w}_3^*)^T H \nabla \mathcal{L}(\boldsymbol{w}_3^*) \leq 0. \tag{24}$$

Since we assumed that $\mathcal{L}$ is convex and $\mathcal{L}(\boldsymbol{w}_3^*)$ is not a minimum ($\mathcal{L}(\boldsymbol{w}_3^*) > \mathcal{L}(\boldsymbol{w}_1)$), we have that $\nabla\mathcal{L}(\boldsymbol{w}_3^*) \neq 0$. Therefore, equation 24 contradicts with $\mathcal{L}$ being strictly convex, and we have $F(\mathcal{L}(\boldsymbol{w}_1)) < F(\mathcal{L}(\boldsymbol{w}_2))$.

We have shown that $\mathcal{L}(\boldsymbol{w}_1) < \mathcal{L}(\boldsymbol{w}_2)$ implies $F(\mathcal{L}(\boldsymbol{w}_1)) < F(\mathcal{L}(\boldsymbol{w}_2))$. Taking the contrapositive and switching $\boldsymbol{w}_1$ and $\boldsymbol{w}_2$, $F(\mathcal{L}(\boldsymbol{w}_1)) \leq F(\mathcal{L}(\boldsymbol{w}_2))$ implies $\mathcal{L}(\boldsymbol{w}_1) \leq \mathcal{L}(\boldsymbol{w}_2)$. Equivalently, $\max_{g \in G}\|\nabla\mathcal{L}(g \cdot \boldsymbol{w}_1)\|^2 \leq \max_{g \in G}\|\nabla\mathcal{L}(g \cdot \boldsymbol{w}_2)\|^2$ implies that $\mathcal{L}(\boldsymbol{w}_1) \leq \mathcal{L}(\boldsymbol{w}_2)$.

Finally, since

$$\max_{g \in G}\|\nabla\mathcal{L}(g \cdot \boldsymbol{w}_1)\|^2 = \|\nabla\mathcal{L}(\boldsymbol{w}_2)\|^2 \leq \max_{g \in G}\|\nabla\mathcal{L}(g \cdot \boldsymbol{w}_2)\|^2, \tag{25}$$

we have $\mathcal{L}(\boldsymbol{w}_1) \leq \mathcal{L}(\boldsymbol{w}_2)$. $\qquad\square$

## B  TELEPORTATION AND NEWTON'S METHOD

**Lemma B.1** (One step of Newton's Method). *Let $f(x)$ be a $\mu$–strongly convex and $L$–smooth function, that is, we have a global lower bound on the Hessian given by*

$$LI \succeq \nabla^2 f(x) \succeq \mu I, \quad \forall x \in \mathbb{R}^n. \tag{26}$$

*Furthermore, if the Hessian is also $G$–Lipschitz*

$$\|\nabla^2 f(x) - \nabla^2 f(y)\| \leq G\|x - y\| \tag{27}$$

*then Newton's method*

$$x^{k+1} = x^k - \lambda_k \nabla^2 f(x^k)^{-1}\nabla f(x^k)$$

*has a mixed linear and quadratic convergence according to*

$$\|x^{k+1} - x^*\| \leq \frac{G}{2\mu}\|x^k - x^*\|^2 + |1 - \lambda_k|\frac{L}{2\mu}\|x^k - x^*\|. \tag{28}$$

*Proof.*

$$x^{k+1} - x^* = x^k - x^* - \lambda_k \nabla^2 f(x^k)^{-1}\left(\nabla f(x^k) - \nabla f(x^*)\right)$$

$$= x^k - x^* - \lambda_k \nabla^2 f(x^k)^{-1} \int_{s=0}^{1} \nabla^2 f(x^k + s(x^* - x^k))(x^k - x^*)ds \quad \text{(Mean value theorem)}$$

$$= \nabla^2 f(x^k)^{-1} \int_{s=0}^{1} \left(\nabla^2 f(x^k) - \lambda_k \nabla^2 f(x^k + s(x^* - x^k))\right)(x^k - x^*)ds$$

$$= \nabla^2 f(x^k)^{-1} \int_{s=0}^{1} \left(\nabla^2 f(x^k) - \nabla^2 f(x^k + s(x^* - x^k))\right.$$

$$\left. + (1 - \lambda_k)\nabla^2 f(x^k + s(x^* - x^k))\right)(x^k - x^*)ds$$

Let $\delta_k := \|x^{k+1} - x^*\|$. Taking norms we have that

$$\delta_{k+1} \leq \|\nabla^2 f(x^k)^{-1}\| \int_{s=0}^{1} \left(\|\nabla^2 f(x^k) - \nabla^2 f(x^k + s(x^* - x^k))\|\right.$$

$$\left. + |1 - \lambda_k|\|\nabla^2 f(x^k + s(x^* - x^k))\|\right)\delta_k ds$$

$$\overset{equation\ 27 + equation\ 26}{\leq} \frac{G}{\mu}\int_{s=0}^{1} s\|x^k - x^*\|^2 ds + |1 - \lambda_k|\frac{L}{\mu}\int_{s=0}^{1} s\|x^k - x^*\|ds$$

$$= \frac{G}{2\mu}\|x^k - x^*\|^2 + |1 - \lambda_k|\frac{L}{2\mu}\|x^k - x^*\|.$$

$\qquad\square$

The assumptions on for this proof can be relaxed, since we only require the Hessian is Lipschitz and lower bounded in a $\frac{\mu}{2L}$–ball around $x^*$.

**Proposition 3.2** (Quadratic term in convergence rate). *Let $\mathcal{L}$ be strictly convex and let $w_0 \in \mathbb{R}^d$. Let*

$$w' \in \arg\max_{w \in \mathbb{R}^d} \frac{1}{2}\|\nabla\mathcal{L}(w)\|^2 \quad \textit{subject to} \quad \mathcal{L}(w) = \mathcal{L}(w_0). \tag{29}$$

*If $\nabla\mathcal{L}(w') \neq 0$ then there exists $\lambda_0$ such that*

$$0 \leq \lambda_0 \leq \lambda_{\max}(\nabla^2\mathcal{L}(w_0))$$

*and one step of gradient descent with learning rate $\gamma > 0$ gives*

$$\begin{aligned} w_1 &= w' - \gamma\nabla\mathcal{L}(w') \\ &= w' - \gamma\lambda_0\nabla^2\mathcal{L}(w')^{-1}\nabla\mathcal{L}(w'). \end{aligned} \tag{30}$$

*Consequently, letting $w' = g_0 \circ w_0$, and if $\gamma \leq \frac{1}{\lambda_0}$ then under the assumptions of Lemma B.1 we have that*

$$\|w_1 - w^*\| \leq \frac{G}{2\mu}\|g_0 \circ w_0 - x^*\|^2 + |1 - \gamma\lambda_0|\frac{L}{2\mu}\|g_0 \circ w_0 - w^*\|.$$

*Proof.* The Lagrangian associated to equation 29 is given by

$$L(w, \lambda) = \frac{1}{2}\|\nabla\mathcal{L}(w)\|^2 + \lambda(\mathcal{L}(w_0) - \mathcal{L}(w)).$$

Taking the derivative in $w$ and setting it to zero gives

$$\nabla_w L(w, \lambda_0) = 0 \implies \nabla^2\mathcal{L}(w)\nabla\mathcal{L}(w) - \lambda_0\nabla\mathcal{L}(w) = 0. \tag{31}$$

Re-arranging we have that

$$\nabla\mathcal{L}(w) = \lambda_0\nabla^2\mathcal{L}(w)^{-1}\nabla\mathcal{L}(w).$$

If $\nabla\mathcal{L}(w') \neq 0$ then from the above we have that

$$\|\nabla\mathcal{L}(w)\|^2 = \lambda_0\nabla\mathcal{L}(w)^\top\nabla^2\mathcal{L}(w)^{-1}\nabla\mathcal{L}(w) > 0.$$

Since $\nabla^2\mathcal{L}(w)^{-1}$ is positive definite we have that $\nabla\mathcal{L}(w)^\top\nabla^2\mathcal{L}(w)^{-1}\nabla\mathcal{L}(w) \geq 0$, and consequently $\lambda_0 > 0$. Finally from equation 31 we have that $\lambda_0$ is an eigenvalue of $\nabla^2\mathcal{L}(w)$ and thus it must be smaller or equal to the largest eigenvalue of $\nabla^2\mathcal{L}(w)$.

$\square$

## C   IS ONE TELEPORTATION ENOUGH TO FIND THE OPTIMAL TRAJECTORY?

This section contains proofs for the results in Section 3.3. For readability, we repeat some of the definitions here.

Consider the parameter space $\mathcal{M} = \mathbb{R}^n$. Let $V : \mathbb{R}^n \to T\mathbb{R}^n$ be a vector field on $\mathbb{R}^n$, where $T\mathbb{R}^n$ denotes the associated tangent bundle. We will write $V = v^i\frac{\partial}{\partial w^i}$ using the component functions $v^i : \mathbb{R}^n \to \mathbb{R}$ and coordinates $w^i$.

Let $\mathcal{L} : \mathcal{M} \to \mathbb{R}$ be a smooth loss function. Let $G$ be a symmetry group of $\mathcal{L}$, i.e. $\mathcal{L}(g \cdot \boldsymbol{w}) = \mathcal{L}(\boldsymbol{w})$ for all $\boldsymbol{w} \in \mathcal{M}$ and $g \in G$. Let $\mathfrak{X}$ be the set of all vector fields on $\mathcal{M}$. Let $R = r^i\frac{\partial}{\partial w^i}$, where $r^i = -\frac{\partial\mathcal{L}}{\partial w_i}$, be the reverse gradient vector field. Let $\mathfrak{X}_\perp = \{A = a^i\frac{\partial}{\partial w^i} \in \mathfrak{X}|\ a^i \in C^\infty(\mathcal{M})$ and $\sum_i a^i(\boldsymbol{w})r^i(\boldsymbol{w}) = 0, \forall \boldsymbol{w} \in \mathcal{M}\}$ be the set of vector fields orthogonal to $R$. If $G$ is a Lie group, the infinitesimal action of its Lie algebra $\mathfrak{g}$ defines a set of vector fields $\mathfrak{X}_\mathfrak{g} \subseteq \mathfrak{X}_\perp$.

A gradient flow is a curve $\gamma : \mathbb{R} \to \mathcal{M}$ where the velocity is the value of $R$ at each point, i.e. $\gamma'(t) = R_{\gamma(t)}$ for all $t \in \mathbb{R}$. The Lie bracket $[A, R]$ defines the derivative of $R$ with respect to $A$. To simplify notation, we write $([W, R]\mathcal{L})(\boldsymbol{w}) = 0$ for a set of vector fields $W \subseteq \mathfrak{X}$ when $([A, R]\mathcal{L})(\boldsymbol{w}) = 0$ for all $A \in W$.

**Proposition 3.4.** *A point $\boldsymbol{w} \in M$ is optimal in a set of vector fields $W$ if and only if $[A, R]\mathcal{L}(\boldsymbol{w}) = 0$ for all $A \in W$.*

*Proof.* Note that $A\mathcal{L} = a^i \frac{\partial \mathcal{L}}{\partial w^i} = 0$. We have

$$[A, R]\mathcal{L} = AR\mathcal{L} - RA\mathcal{L} = A\left(r^i \frac{\partial \mathcal{L}}{\partial w_i}\right) - 0 = -A \left\|\frac{\partial \mathcal{L}}{\partial \boldsymbol{w}}\right\|_2^2 = -Af. \tag{32}$$

The result then follows from Definition 3.3. □

**Proposition 3.5.** *Let $W \subseteq \mathfrak{X}_\perp$ be a set of vector fields that are orthogonal to the gradient of $\mathcal{L}$. If $[A, R]\mathcal{L}(\boldsymbol{w}) = 0$ for all $A \in W$ implies that $R([A, R]\mathcal{L})(\boldsymbol{w}) = 0$ for all $A \in W$, then the gradient flow starting at an optimal point in $W$ is optimal in $W$.*

*Proof.* Consider the gradient flow $\gamma$ that starts at an optimal point in $W$. The derivative of $[A, R]\mathcal{L}$ along $\gamma$ is

$$\frac{d}{dt}[A, R]\mathcal{L}(\gamma(t)) = \gamma'(t)([A, R]\mathcal{L})(\gamma(t)) = -R[A, R]\mathcal{L}(\gamma(t)). \tag{33}$$

Since $\gamma(0)$ is an optimal point, $[A, R]\mathcal{L}(\gamma(0)) = 0$ for all $A \in W$ by Proposition 3.4. By assumption, if $[A, R]\mathcal{L}(\gamma(t)) = 0$ for all $A \in W$, then $R([A, R]\mathcal{L})(\gamma(t)) = 0$ for all $A \in W$. Therefore, both the value and the derivative of $[A, R]\mathcal{L}$ stay 0 along $\gamma$. Since $[A, R]\mathcal{L}(\gamma(t)) = 0$ for all $t \in \mathbb{R}$, $\gamma$ is optimal in $W$. □

To help check when Proposition 3.5 is satisfied, we provide an alternative form of $R[A, R]\mathcal{L}(\boldsymbol{w})$ under the assumption that $[A, R]\mathcal{L}(\boldsymbol{w}) = 0$. We will use the following lemmas in the proof.

**Lemma C.1.** *For two vectors $\boldsymbol{v}, \boldsymbol{w} \in \mathbb{R}^n$, if $\boldsymbol{v}^T \boldsymbol{w} = 0$ and $\boldsymbol{w} \neq \boldsymbol{0}$, then there exists an anti-symmetric matrix $M \in \mathbb{R}^{n \times n}$ such that $\boldsymbol{v} = M\boldsymbol{w}$.*

*Proof.* Let $\boldsymbol{w}_0 = [1, 0, ..., 0]^T \in \mathbb{R}^n$. Consider a list of $n-1$ anti-symmetric matrices $M_i \in \mathbb{R}^{n \times n}$, where

$$M_{ij}^k = \begin{cases} -1, & \text{if } j = 1 \text{ and } k = i+1 \\ 1, & \text{if } j = i+1 \text{ and } k = 1 \\ 0, & \text{otherwise} \end{cases} \tag{34}$$

In matrix form, the $M_i$'s are

$$M_1 = \begin{bmatrix} 0 & -1 & 0 & ... & 0 \\ 1 & 0 & 0 & ... & 0 \\ 0 & 0 & 0 & ... & 0 \\ & & ... & & \\ 0 & 0 & 0 & ... & 0 \end{bmatrix}, M_2 = \begin{bmatrix} 0 & 0 & -1 & ... & 0 \\ 0 & 0 & 0 & ... & 0 \\ 1 & 0 & 0 & ... & 0 \\ & & ... & & \\ 0 & 0 & 0 & ... & 0 \end{bmatrix}, ..., M_{n-1} = \begin{bmatrix} 0 & 0 & 0 & ... & -1 \\ 0 & 0 & 0 & ... & 0 \\ 0 & 0 & 0 & ... & 0 \\ & & ... & & \\ 1 & 0 & 0 & ... & 0 \end{bmatrix}. \tag{35}$$

Since $M_i$'s are anti-symmetric, $M_i \boldsymbol{w}_0$ is orthogonal to $\boldsymbol{w}_0$. The norm of $M_i \boldsymbol{w}_0 = \mathbf{e}_{i+1}$ is 1. Additionally, $M_i \boldsymbol{w}_0$ is orthogonal to $M_j \boldsymbol{w}_0$ for $i \neq j$:

$$(M_i \boldsymbol{w}_0)^T (M_j \boldsymbol{w}_0) = \mathbf{e}_{i+1}^T \mathbf{e}_{j+1} = \delta_{ij}. \tag{36}$$

Denote $\boldsymbol{w}_0^\perp = \{\boldsymbol{x} \in \mathbb{R}^n : \boldsymbol{x}^T \boldsymbol{w}_0 = 0\}$ as the orthogonal complement of $\boldsymbol{w}_0$. Then $M_i \boldsymbol{w}_0$ forms a basis of $\boldsymbol{w}_0^\perp$. Next, we extend this to an arbitrary $\boldsymbol{w} \in \mathbb{R}^n$.

Let $\hat{\boldsymbol{w}} = \frac{\boldsymbol{w}}{\|\boldsymbol{w}\|_2}$. Since $\hat{\boldsymbol{w}}$ has norm 1, there exists an orthogonal matrix $R$ such that $\hat{\boldsymbol{w}} = R\boldsymbol{w}_0$. Let $M_i' = RM_iR^T$. Then $M_i'$ is anti-symmetric:

$$(RM_iR^T)^T = RM_i^T R^T = -RM_iR^T. \tag{37}$$

It follows that $M_i' \hat{\boldsymbol{w}}$ is orthogonal to $\hat{\boldsymbol{w}}$. The norm of $M_i' \hat{\boldsymbol{w}}$ is $\|(RM_iR^T)(R\boldsymbol{w}_0)\| = \|RM_i\boldsymbol{w}_0\| = \|M_i\boldsymbol{w}_0\| = 1$. Additionally, $M_i'\hat{\boldsymbol{w}}$ is orthogonal to $M_j'\hat{\boldsymbol{w}}$ for $i \neq j$:

$$\begin{aligned} (M_i'\hat{\boldsymbol{w}})^T (M_j'\hat{\boldsymbol{w}}) &= (RM_iR^T R\boldsymbol{w}_0)^T (RM_jR^T R\boldsymbol{w}_0) \\ &= \boldsymbol{w}_0^T R^T RM_i^T R^T RM_jR^T R\boldsymbol{w}_0 \\ &= \boldsymbol{w}_0^T M_i^T M_j \boldsymbol{w}_0 \\ &= \delta_{ij}. \end{aligned} \tag{38}$$

Therefore, $M_i'\hat{\boldsymbol{w}}$ spans $\hat{\boldsymbol{w}}^\perp = \boldsymbol{w}^\perp$. This means that any vector $\boldsymbol{v} \in \boldsymbol{w}^\perp$ can be written as a linear combination of $M_i'\hat{\boldsymbol{w}}$. That is, there exists $k_1, ..., k_n \in \mathbb{R}$, such that $\boldsymbol{v} = \sum_i k_i(M_i'\hat{\boldsymbol{w}})$. To find the anti-symmetric $M$ that takes $\boldsymbol{w}$ to $\boldsymbol{v}$, note that

$$\boldsymbol{v} = \left(\sum_i k_i M_i'\right)\hat{\boldsymbol{w}} = \left(\|\boldsymbol{w}\|_2^{-1}\sum_i k_i M_i'\right)\boldsymbol{w}. \tag{39}$$

Since the sum of anti-symmetric matrices is anti-symmetric, and the product of an anti-symmetric matrix and a scalar is also anti-symmetric, $\|\boldsymbol{w}\|_2^{-1}\sum_i k_i M_i'$ is anti-symmetric. $\qquad\square$

**Lemma C.2.** *Let $\boldsymbol{v} \in \mathbb{R}^n$ be a nonzero vector. Then the two sets $\{M\boldsymbol{v} : M \in \mathbb{R}^{n\times n}, M^T = -M\}$ and $\{\boldsymbol{w} \in \mathbb{R}^n : \boldsymbol{w}^T\boldsymbol{v} = 0\}$ are equal.*

*Proof.* Let $A = \{M\boldsymbol{v} : M \in \mathbb{R}^{n\times n}, M^T = M^{-1}\}$ and $B = \{\boldsymbol{w} \in \mathbb{R}^n : \boldsymbol{w}^T\boldsymbol{v} = 0\}$. Since $(M\boldsymbol{v})^T\boldsymbol{v} = 0$ for all anti-symmetric $M$, every element in $A$ is in $B$. By Lemma C.1, every element in $B$ is in $A$. Therefore $A = B$. $\qquad\square$

Let $S = \{(M\frac{\partial\mathcal{L}}{\partial\boldsymbol{w}})^i\frac{\partial}{\partial w^i} \in \mathfrak{X}| M \in \mathbb{R}^{n\times n}, M^T = -M\}$ be the set of vector fields constructed by multiplying the gradient by an anti-symmetric matrix. Recall that $R = -\frac{\partial\mathcal{L}}{\partial w_i}\frac{\partial}{\partial w^i}$ is the reverse gradient vector field, and $\mathfrak{X}_\perp = \{a^i\frac{\partial}{\partial w^i}| \sum_i a^i(\boldsymbol{w})\frac{\partial\mathcal{L}(\boldsymbol{w})}{\partial w^i} = 0, \forall \boldsymbol{w} \in \mathcal{M}\}$ is the set of all vector fields orthogonal to $R$. From Lemma C.2, we have $S = \mathfrak{X}_\perp$. Therefore, a point $\boldsymbol{w}$ is an optimal point in $S$ if and only if $\boldsymbol{w}$ is an optimal point in $\mathfrak{X}_\perp$.

We are now ready to prove the following proposition, which provides another way to check the condition in Proposition 3.5.

**Proposition 3.6.** *If at all optimal points in $S$,*

$$M_\alpha^j \frac{\partial\mathcal{L}}{\partial w_k}\frac{\partial\mathcal{L}}{\partial w_\alpha}\frac{\partial^3\mathcal{L}}{\partial w^k\partial w_i\partial w^j}\frac{\partial\mathcal{L}}{\partial w^i} = 0 \tag{40}$$

*for all anti-symmetric matrix $M \in \mathbb{R}^{n\times n}$, then the gradient flow starting at an optimal point in $S$ is optimal in $S$.*

*Proof.* Expanding $R[A, R]\mathcal{L}$, we have

$$R[A, R]\mathcal{L} = R\left(A\left(r^i\frac{\partial\mathcal{L}}{\partial w^i}\right) - 0\right)$$
$$= r^k\frac{\partial}{\partial w^k}\left(a^j\frac{\partial}{\partial w_j}\left(r^i\frac{\partial\mathcal{L}}{\partial w^i}\right)\right)$$
$$= r^k\frac{\partial}{\partial w^k}\left(a^j\left(\frac{\partial r^i}{\partial w^j}\frac{\partial\mathcal{L}}{\partial w^i} + r^i\frac{\partial}{\partial w^j}\frac{\partial\mathcal{L}}{\partial w^i}\right)\right)$$
$$= -r^k\frac{\partial}{\partial w^k}\left(a^j\left(\left(\frac{\partial}{\partial w^j}\frac{\partial\mathcal{L}}{\partial w_i}\right)\frac{\partial\mathcal{L}}{\partial w^i} + \frac{\partial\mathcal{L}}{\partial w_i}\frac{\partial}{\partial w^j}\frac{\partial\mathcal{L}}{\partial w^i}\right)\right)$$
$$= -2r^k\frac{\partial}{\partial w^k}\left(a^j\frac{\partial^2\mathcal{L}}{\partial w_i\partial w^j}\frac{\partial\mathcal{L}}{\partial w^i}\right)$$
$$= -2r^k\left(\frac{\partial a^j}{\partial w^k}\frac{\partial^2\mathcal{L}}{\partial w_i\partial w^j}\frac{\partial\mathcal{L}}{\partial w^i} + a^j\frac{\partial}{\partial w^k}\left(\frac{\partial^2\mathcal{L}}{\partial w_i\partial w^j}\frac{\partial\mathcal{L}}{\partial w^i}\right)\right)$$
$$= 2\frac{\partial\mathcal{L}}{\partial w_k}\frac{\partial a^j}{\partial w^k}\frac{\partial^2\mathcal{L}}{\partial w_i\partial w^j}\frac{\partial\mathcal{L}}{\partial w^i} + 2\frac{\partial\mathcal{L}}{\partial w_k}a^j\frac{\partial}{\partial w^k}\left(\frac{\partial^2\mathcal{L}}{\partial w_i\partial w^j}\frac{\partial\mathcal{L}}{\partial w^i}\right) \tag{41}$$

Assume that $\boldsymbol{w}$ is an optimal point in $S$. By Lemma C.2, $\boldsymbol{w}$ is also an optimal point in $\mathfrak{X}_\perp$. By Lemma C.4 in Zhao et al. (2022), $\frac{\partial\mathcal{L}}{\partial\boldsymbol{w}}$ is an eigenvector of $\frac{\partial^2\mathcal{L}}{\partial w_i\partial w^j}$. Therefore, $\frac{\partial^2\mathcal{L}}{\partial w_i\partial w^j}\frac{\partial\mathcal{L}}{\partial w^i} = \lambda\frac{\partial\mathcal{L}}{\partial w^j}$ for some $\lambda \in \mathbb{C}$. Additionally, $a^j = M_\alpha^j\frac{\partial\mathcal{L}}{\partial w_\alpha}$ and $\frac{\partial a^j}{\partial w^k} = M_\alpha^j\frac{\partial^2\mathcal{L}}{\partial w_\alpha\partial w^k}$. We are now ready to simplify both terms in equation 41.

For the first term in equation 41,

$$
\begin{aligned}
\frac{\partial \mathcal{L}}{\partial w_k} \frac{\partial a^j}{\partial w^k} \frac{\partial^2 \mathcal{L}}{\partial w_i \partial w^j} \frac{\partial \mathcal{L}}{\partial w^i} &= \frac{\partial \mathcal{L}}{\partial w_k} M_\alpha^j \frac{\partial^2 \mathcal{L}}{\partial w_\alpha \partial w^k} \frac{\partial^2 \mathcal{L}}{\partial w_i \partial w^j} \frac{\partial \mathcal{L}}{\partial w^i} \\
&= M_\alpha^j \left( \frac{\partial^2 \mathcal{L}}{\partial w_\alpha \partial w^k} \frac{\partial \mathcal{L}}{\partial w_k} \right) \left( \frac{\partial^2 \mathcal{L}}{\partial w_i \partial w^j} \frac{\partial \mathcal{L}}{\partial w^i} \right) \\
&= M_\alpha^j \left( \lambda_1 \frac{\partial \mathcal{L}}{\partial w_\alpha} \right) \left( \lambda_2 \frac{\partial \mathcal{L}}{\partial w^j} \right) \\
&= \lambda_1 \lambda_2 M_\alpha^j \frac{\partial \mathcal{L}}{\partial w_\alpha} \frac{\partial \mathcal{L}}{\partial w^j} \\
&= 0
\end{aligned}
\tag{42}
$$

The last equality holds because $M$ is anti-symmetric.

For the second term in equation 41,

$$
\begin{aligned}
\frac{\partial \mathcal{L}}{\partial w_k} a^j \frac{\partial}{\partial w^k} \left( \frac{\partial^2 \mathcal{L}}{\partial w_i \partial w^j} \frac{\partial \mathcal{L}}{\partial w^i} \right) &= \frac{\partial \mathcal{L}}{\partial w_k} a^j \left( \frac{\partial^3 \mathcal{L}}{\partial w^k \partial w_i \partial w^j} \frac{\partial \mathcal{L}}{\partial w^i} + \frac{\partial^2 \mathcal{L}}{\partial w_i \partial w^j} \frac{\partial^2 \mathcal{L}}{\partial w^k \partial w^i} \right) \\
&= \frac{\partial \mathcal{L}}{\partial w_k} M_\alpha^j \frac{\partial \mathcal{L}}{\partial w_\alpha} \left( \frac{\partial^3 \mathcal{L}}{\partial w^k \partial w_i \partial w^j} \frac{\partial \mathcal{L}}{\partial w^i} + \frac{\partial^2 \mathcal{L}}{\partial w_i \partial w^j} \frac{\partial^2 \mathcal{L}}{\partial w^k \partial w^i} \right) \\
&= M_\alpha^j \frac{\partial \mathcal{L}}{\partial w_k} \frac{\partial \mathcal{L}}{\partial w_\alpha} \frac{\partial^3 \mathcal{L}}{\partial w^k \partial w_i \partial w^j} \frac{\partial \mathcal{L}}{\partial w^i} + \lambda_1 \lambda_2 M_\alpha^j \frac{\partial \mathcal{L}}{\partial w_\alpha} \frac{\partial \mathcal{L}}{\partial w^j} \\
&= M_\alpha^j \frac{\partial \mathcal{L}}{\partial w_k} \frac{\partial \mathcal{L}}{\partial w_\alpha} \frac{\partial^3 \mathcal{L}}{\partial w^k \partial w_i \partial w^j} \frac{\partial \mathcal{L}}{\partial w^i}
\end{aligned}
\tag{43}
$$

In summary,

$$
R[A, R]\mathcal{L} = 2 M_\alpha^j \frac{\partial \mathcal{L}}{\partial w_k} \frac{\partial \mathcal{L}}{\partial w_\alpha} \frac{\partial^3 \mathcal{L}}{\partial w^k \partial w_i \partial w^j} \frac{\partial \mathcal{L}}{\partial w^i}.
\tag{44}
$$

Since we assumed that $[A, R]\mathcal{L}(\boldsymbol{w}) = 0$, when $R[A, R]\mathcal{L}(\boldsymbol{w}) = 0$ for all $A \in S$, the gradient flow starting at an optimal point in $S$ is optimal in $S$. $\qquad\square$

**Proposition C.3.** *If $\frac{\partial^3 \mathcal{L}}{\partial w^k \partial w^i \partial w^j} \frac{\partial \mathcal{L}}{\partial w^\alpha} = \frac{\partial^3 \mathcal{L}}{\partial w^k \partial w^i \partial w^\alpha} \frac{\partial \mathcal{L}}{\partial w^j}$ holds for all $i, k, j, \alpha$, then $M_\alpha^j \frac{\partial \mathcal{L}}{\partial w_k} \frac{\partial \mathcal{L}}{\partial w_\alpha} \frac{\partial^3 \mathcal{L}}{\partial w^k \partial w_i \partial w^j} \frac{\partial \mathcal{L}}{\partial w^i} = 0$ holds for all anti-symmetric matrices $M \in \mathbb{R}^{n \times n}$.*

*Proof.* If $\frac{\partial^3 \mathcal{L}}{\partial w^k \partial w^i \partial w^j} \frac{\partial \mathcal{L}}{\partial w^\alpha} = \frac{\partial^3 \mathcal{L}}{\partial w^k \partial w^i \partial w^\alpha} \frac{\partial \mathcal{L}}{\partial w^j}$ for all $i, k, j, \alpha$, then

$$
\begin{aligned}
& M_\alpha^j \frac{\partial \mathcal{L}}{\partial w_k} \frac{\partial \mathcal{L}}{\partial w_\alpha} \frac{\partial^3 \mathcal{L}}{\partial w^k \partial w_i \partial w^j} \frac{\partial \mathcal{L}}{\partial w^i} \\
&= \sum_{i,k,\alpha<j} M_\alpha^j \frac{\partial \mathcal{L}}{\partial w_k} \frac{\partial \mathcal{L}}{\partial w_\alpha} \frac{\partial^3 \mathcal{L}}{\partial w^k \partial w_i \partial w^j} \frac{\partial \mathcal{L}}{\partial w^i} + \sum_{i,k,\alpha>j} M_\alpha^j \frac{\partial \mathcal{L}}{\partial w_k} \frac{\partial \mathcal{L}}{\partial w_\alpha} \frac{\partial^3 \mathcal{L}}{\partial w^k \partial w_i \partial w^j} \frac{\partial \mathcal{L}}{\partial w^i} \\
&= \sum_{i,k,\alpha<j} M_\alpha^j \frac{\partial \mathcal{L}}{\partial w_k} \frac{\partial \mathcal{L}}{\partial w_\alpha} \frac{\partial^3 \mathcal{L}}{\partial w^k \partial w_i \partial w^j} \frac{\partial \mathcal{L}}{\partial w^i} + \sum_{i,k,j>\alpha} M_j^\alpha \frac{\partial \mathcal{L}}{\partial w_k} \frac{\partial \mathcal{L}}{\partial w_j} \frac{\partial^3 \mathcal{L}}{\partial w^k \partial w_i \partial w^\alpha} \frac{\partial \mathcal{L}}{\partial w^i} \\
&= \sum_{i,k,\alpha<j} M_\alpha^j \frac{\partial \mathcal{L}}{\partial w_k} \frac{\partial \mathcal{L}}{\partial w_\alpha} \frac{\partial^3 \mathcal{L}}{\partial w^k \partial w_i \partial w^j} \frac{\partial \mathcal{L}}{\partial w^i} + \sum_{i,k,j>\alpha} -M_\alpha^j \frac{\partial \mathcal{L}}{\partial w_k} \frac{\partial \mathcal{L}}{\partial w_j} \frac{\partial^3 \mathcal{L}}{\partial w^k \partial w_i \partial w^\alpha} \frac{\partial \mathcal{L}}{\partial w^i} \\
&= \sum_{i,k,\alpha<j} M_\alpha^j \frac{\partial \mathcal{L}}{\partial w_k} \frac{\partial \mathcal{L}}{\partial w^i} \left( \frac{\partial \mathcal{L}}{\partial w_\alpha} \frac{\partial^3 \mathcal{L}}{\partial w^k \partial w_i \partial w^j} - \frac{\partial \mathcal{L}}{\partial w_j} \frac{\partial^3 \mathcal{L}}{\partial w^k \partial w_i \partial w^\alpha} \right) \\
&= 0,
\end{aligned}
\tag{45}
$$

where the first equality uses that the diagonal of an anti-symmetric matrix is 0, the second equality swaps $\alpha$ and $j$ in the second term, the third equality uses that $M$ is anti-symmetric. $\qquad\square$

**Example (Quadratic function)**   Consider the quadratic function $\mathcal{L}(\boldsymbol{w}) = \frac{1}{2}\boldsymbol{w}^T A \boldsymbol{w} + \mathbf{b}^T \boldsymbol{w} + \mathbf{c}$, where $A \in \mathbb{R}^{n \times n}$ is symmetric, $\mathbf{b}, \mathbf{c} \in \mathbb{R}^n$, and $\boldsymbol{w} \in \mathbb{R}^n$. Two examples of quadratic functions are the ellipse $\mathcal{L}_e(w_1, w_2) = \frac{1}{2}(w_1^2 + \lambda^2 w_2^2)$ and the Booth function $\mathcal{L}_b(w_1, w_2) = (w_1 + 2w_2 - 7)^2 + (2w_1 + w_2 - 5)^2$. Since the third derivative of $\mathcal{L}$ is 0, one teleportation guarantees optimal trajectory.

## D  GROUP ACTIONS AND CURVES ON MINIMA

### D.1  GROUP ACTIONS FOR MLP

Consider a multi-layer neural network with elementwise activation function $\sigma$. The output of the $m^{th}$ layer is $h_m = \sigma(W_m h_{m-1})$, where $W_m \in \mathbb{R}^{d_m \times d_{m-1}}$ is the weight, $h_{m-1} \in \mathbb{R}^{d_{m-1} \times k}$ is the output of the $m - 1^{th}$ layer, and $h_0 \in \mathbb{R}^{d_0 \times k}$ is the data.

Assuming that $\sigma\left(g_m W_{m-1} h_{m-2}\right)$ is invertible, for $g_m \in \mathrm{GL}_{d_{m-1}}(\mathbb{R})$, the following transformation is a loss-preserving group action:

$$g_m \cdot W_k = \begin{cases} W_m \sigma\left(W_{m-1} h_{m-2}\right) \sigma\left(g_m W_{m-1} h_{m-2}\right)^{-1} & k = m \\ g_m W_{m-1} & k = m - 1 \\ W_k & k \notin \{m, m-1\} \end{cases} \tag{46}$$

Usually, the assumption does not hold (Zhao et al., 2023). Hence the above transformation may not preserve loss or be a valid group action. Nevertheless, we observe in practice that the change in the loss value is often small after such transformations on parameters. We therefore refer to equation (46) as an approximate symmetry and adopt it in the teleportation algorithm. Due to the possibility that $\sigma\left(g_m W_{m-1} h_{m-2}\right)$ is not invertible, we use pseudoinverses in implementations.

### D.2  CURVATURE

The curvature of a curve $\gamma : \mathbb{R} \to \mathbb{R}^n$ is $\kappa(t) = \frac{\|T'(t)\|}{\|\gamma'(t)\|}$, where $T(t) = \frac{\gamma'(t)}{\|\gamma'(t)\|}$ is the unit tangent vector. The curvature can be written as a function of $\gamma'$ and $\gamma''$ (Aléssio, 2012; Shelekhov, 2021):

$$\kappa(t) = \frac{\left[\|\gamma'\|^2 \|\gamma''\|^2 - (\gamma' \cdot \gamma'')^2\right]^{\frac{1}{2}}}{\|\gamma'\|^3}. \tag{47}$$

### D.3  THE DERIVATIVE OF CURVATURE

To compute the derivative of $\kappa(t)$, we first list the derivatives of a few commonly used terms:

$$\frac{d}{dt}\|\gamma'\|^2 = \frac{d}{dt}(\gamma_1'^2 + \gamma_2'^2 + \gamma_3'^2 + ...) = 2\gamma_1'\gamma_1'' + 2\gamma_2'\gamma_2'' + 2\gamma_3'\gamma_3'' + ... = 2\gamma' \cdot \gamma''$$

$$\frac{d}{dt}\|\gamma''\|^2 = \frac{d}{dt}(\gamma_1''^2 + \gamma_2''^2 + \gamma_3''^2 + ...) = 2\gamma_1''\gamma_1''' + 2\gamma_2''\gamma_2''' + 2\gamma_3''\gamma_3''' + ... = 2\gamma'' \cdot \gamma'''$$

$$\frac{d}{dt}(\gamma' \cdot \gamma'') = \frac{d}{dt}(\gamma_1'\gamma_1'' + \gamma_2'\gamma_2'' + \gamma_3'\gamma_3''...) = \gamma_1'\gamma_1''' + \gamma_1''\gamma_1'' + ... = \|\gamma''\|^2 + \gamma' \cdot \gamma''' \tag{48}$$

The derivatives of the numerator and denominator of $\kappa$ are:

$$\frac{d}{dt}\left[\|\gamma'\|^2\|\gamma''\|^2 - (\gamma' \cdot \gamma'')^2\right]^{\frac{1}{2}} = \frac{1}{2}\left[\|\gamma'\|^2\|\gamma''\|^2 - (\gamma' \cdot \gamma'')^2\right]^{-\frac{1}{2}} \frac{d}{dt}\left[\|\gamma'\|^2\|\gamma''\|^2 - (\gamma' \cdot \gamma'')^2\right]$$

$$= \frac{1}{2}\left[\|\gamma'\|^2\|\gamma''\|^2 - (\gamma' \cdot \gamma'')^2\right]^{-\frac{1}{2}}$$

$$\left[\|\gamma'\|^2\frac{d}{dt}\|\gamma''\|^2 + \|\gamma''\|^2\frac{d}{dt}\|\gamma'\|^2 - 2(\gamma' \cdot \gamma'')\frac{d}{dt}(\gamma' \cdot \gamma'')\right]$$

$$= \frac{1}{2}\left[\|\gamma'\|^2\|\gamma''\|^2 - (\gamma' \cdot \gamma'')^2\right]^{-\frac{1}{2}}$$

$$\left[2\|\gamma'\|^2(\gamma'' \cdot \gamma''') + 2\|\gamma''\|^2(\gamma' \cdot \gamma'') - 2(\gamma' \cdot \gamma'')(\|\gamma''\|^2 + \gamma' \cdot \gamma''')\right]$$

$$= \left[\|\gamma'\|^2\|\gamma''\|^2 - (\gamma' \cdot \gamma'')^2\right]^{-\frac{1}{2}}\left[\|\gamma'\|^2(\gamma'' \cdot \gamma''') - (\gamma' \cdot \gamma'')(\gamma' \cdot \gamma''')\right],$$

$$\tag{49}$$

and

$$\frac{d}{dt}\|\gamma'\|^3 = \frac{d}{dt}(\|\gamma'\|^2)^{\frac{3}{2}} = \frac{3}{2}(\|\gamma'\|^2)^{\frac{1}{2}}\frac{d}{dt}\|\gamma'\|^2 = \frac{3}{2}(\|\gamma'\|^2)^{\frac{1}{2}}(2\gamma' \cdot \gamma'') = 3\|\gamma'\|(\gamma' \cdot \gamma''). \quad (50)$$

Using the derivatives above, the derivative of $\kappa$ is

$$
\begin{aligned}
\kappa'(t) &= \frac{\left[\frac{d}{dt}\left[\|\gamma'\|^2\|\gamma''\|^2 - (\gamma' \cdot \gamma'')^2\right]^{\frac{1}{2}}\right]\|\gamma'\|^3 - \left[\|\gamma'\|^2\|\gamma''\|^2 - (\gamma' \cdot \gamma'')^2\right]^{\frac{1}{2}}\left[\frac{d}{dt}\|\gamma'\|^3\right]}{\|\gamma'\|^6} \\[2mm]
&= \frac{\left[\|\gamma'\|^2\|\gamma''\|^2 - (\gamma' \cdot \gamma'')^2\right]^{-\frac{1}{2}}\left[\|\gamma'\|^2(\gamma'' \cdot \gamma''') - (\gamma' \cdot \gamma'')(\gamma' \cdot \gamma''')\right]\|\gamma'\|^3}{\|\gamma'\|^6} \\
&\quad - \frac{\left[\|\gamma'\|^2\|\gamma''\|^2 - (\gamma' \cdot \gamma'')^2\right]^{\frac{1}{2}}3\|\gamma'\|(\gamma' \cdot \gamma'')}{\|\gamma'\|^6} \\[2mm]
&= \frac{\left[\|\gamma'\|^2\|\gamma''\|^2 - (\gamma' \cdot \gamma'')^2\right]^{-\frac{1}{2}}\left[\|\gamma'\|^2(\gamma'' \cdot \gamma''') - (\gamma' \cdot \gamma'')(\gamma' \cdot \gamma''')\right]\|\gamma'\|^2}{\|\gamma'\|^5} \\
&\quad - \frac{\left[\|\gamma'\|^2\|\gamma''\|^2 - (\gamma' \cdot \gamma'')^2\right]^{\frac{1}{2}}3(\gamma' \cdot \gamma'')}{\|\gamma'\|^5}.
\end{aligned}
$$
$$(51)$$

### D.4 THE DERIVATIVES OF CURVES ON MINIMA

Consider the curve $\gamma_M : \mathbb{R} \times \mathbb{R}^n \to \mathbb{R}^n$ where $M \in \text{Lie}(G)$ and

$$\gamma_M(t, \boldsymbol{w}) = \exp(tM) \cdot \boldsymbol{w}. \quad (52)$$

In this section, we derive $\gamma'$, $\gamma''$, and $\gamma'''$, which are needed to compute the curvature $\kappa(t)$ and its derivative $\kappa'(t)$. We are interested in $\kappa$ and $\kappa'$ at $\boldsymbol{w}$, or equivalently, at $t = 0$. To find the derivatives of $\gamma$ at $t = 0$, we write the group action in the following form:

$$\gamma(t) = \sum_{n=0}^{\infty} \frac{f(n)}{n!} t^n. \quad (53)$$

By the uniqueness of Taylor polynomial, the derivatives are $\gamma^{(n)}(0) = f(n)$. In the rest of this subsection, we expand the group action to find $f(n)$.

Consider two consecutive layers $U\sigma(VX)$ in a neural network, where $U \in \mathbb{R}^{m \times h}, V \in \mathbb{R}^{h \times n}$ are weights, $X \in \mathbb{R}^{h \times k}$ is the output from the previous layer, and $\sigma$ is an elementwise activation function. Choosing $G = GL_h(\mathbb{R})$, one group action that leaves the output of these two layers unchanged is:

$$g \cdot (U, V, X) = (g \cdot U, g \cdot V, g \cdot X) = (Ug^{-1}, \sigma^{-1}(g\sigma(VX))X^{-1}, X). \quad (54)$$

Let

$$g = \exp(tM) = \sum_{k=0}^{\infty} \frac{1}{k!}(tM)^k, \quad (55)$$

where $M \in \text{Lie}(G)$ is in the Lie algebra of $G$. The action of $g$ yields

$$g \cdot (U, V, X) = (U\exp(-tM), \sigma^{-1}(\exp(tM)\sigma(VX))X^{-1}, X). \quad (56)$$

Next, we expand $\gamma(t) = g \cdot (U, V)$. The Taylor expansion for $g \cdot U$ is

$$
\begin{aligned}
U\exp(-tM) &= U\sum_{k=0}^{\infty} \frac{1}{k!}(-tM)^k \\
&= U - tUM + \frac{t^2}{2!}UM^2 - \frac{t^3}{3!}UM^3 + O(t^4). \quad (57)
\end{aligned}
$$

The Taylor expansion for $g \cdot V$ is

$$
\sigma^{-1}(\exp(tM)\sigma(VX))X^{-1}
$$

$$
=\sigma^{-1}\left(\left(\sum_{k=0}^{\infty}\frac{1}{k!}(tM)^k\right)\sigma(VX)\right)X^{-1}
$$

$$
=\sigma^{-1}\left(\sigma(VX)+\sum_{k=1}^{\infty}\frac{1}{k!}(tM)^k\sigma(VX)\right)X^{-1}
$$

$$
=\left[\sigma^{-1}(\sigma(VX))+\sum_{j=1}^{\infty}\left(\sum_{k=1}^{\infty}\frac{1}{k!}(tM)^k\sigma(VX)\right)^{\odot j}\odot\left.\frac{\partial^j\sigma^{-1}(A)}{\partial A^j}\right|_{A=\sigma(VX)}\right]X^{-1}
$$

$$
=V+\left[\sum_{j=1}^{\infty}\left(\sum_{k=1}^{\infty}\frac{1}{k!}(tM)^k\sigma(VX)\right)^{\odot j}\odot\left.\frac{\partial^j\sigma^{-1}(A)}{\partial A^j}\right|_{A=\sigma(VX)}\right]X^{-1}, \tag{58}
$$

where $\odot$ denotes element-wise product: $(A\odot B)_{mn}=A_{mn}B_{mn}$, and the superscript $^{\odot}$ denotes elementwise power: $(A^{\odot j})_{mn}=(A_{mn})^j$. The Taylor expansion is of each element individually, because $\sigma$ is element-wise.

Since our goal is to find the first 3 derivatives of $\gamma$, we are only interested in the terms up to $t^3$. Letting

$$
\sum_{k=1}^{\infty}\frac{1}{k!}(tM)^k = tM + t^2\frac{M^2}{2} + t^3\frac{M^3}{6} + O(t^4) \tag{59}
$$

and considering only the $j=1,2,3$ terms, we have

$$
\sigma^{-1}(\exp(tM)\sigma(VX))X^{-1}
$$

$$
=V+\left[\sum_{j=1}^{\infty}\left((tM+t^2\frac{M^2}{2}+t^3\frac{M^3}{6})\sigma(VX)\right)^{\odot j}\odot\left.\frac{\partial^j\sigma^{-1}(A)}{\partial A^j}\right|_{A=\sigma(VX)}\right]X^{-1}+O(t^4)
$$

$$
=V+\left[\left((tM+t^2\frac{M^2}{2}+t^3\frac{M^3}{6})\sigma(VX)\right)\odot\left.\frac{\partial\sigma^{-1}(A)}{\partial A}\right|_{A=\sigma(VX)}\right.
$$

$$
+\left((tM+t^2\frac{M^2}{2}+t^3\frac{M^3}{6})\sigma(VX)\right)^{\odot 2}\odot\left.\frac{\partial^2\sigma^{-1}(A)}{\partial A^2}\right|_{A=\sigma(VX)}
$$

$$
\left.+\left((tM+t^2\frac{M^2}{2}+t^3\frac{M^3}{6})\sigma(VX)\right)^{\odot 3}\odot\left.\frac{\partial^3\sigma^{-1}(A)}{\partial A^3}\right|_{A=\sigma(VX)}\right]X^{-1}+O(t^4)
$$

$$
=V+t\left((M\sigma(VX))\odot\frac{1}{\sigma'(VX)}\right)X^{-1}
$$

$$
+\frac{t^2}{2}\left((M^2\sigma(VX))\odot\frac{1}{\sigma'(VX)}-2(M\sigma(VX))^{\odot 2}\odot\frac{\sigma''(VX)}{\sigma'(VX)^3}\right)X^{-1}
$$

$$
+\frac{t^3}{6}\left((M^3\sigma(VX))\odot\frac{1}{\sigma'(VX)}-6(M\sigma(VX))\odot(M^2\sigma(VX))\odot\frac{\sigma''(VX)}{\sigma'(VX)^3}\right.
$$

$$
\left.+6(M\sigma(VX))^{\odot 3}\odot\left.\frac{\partial^3\sigma^{-1}(A)}{\partial A^3}\right|_{A=\sigma(VX)}\right)X^{-1}
$$

$$
+O(t^4). \tag{60}
$$

Matching terms in equation 57 and equation 60 with equation 53, we have the expressions for $\gamma'$, $\gamma''$, and $\gamma'''$. This allows us to compute the curvature and its derivative using equation 47 and equation 51.

# E  SHARPNESS, CURVATURES, AND THEIR RELATION TO GENERALIZATION

## E.1  ALTERNATIVE DEFINITIONS OF SHARPNESS

A common definition of flat minimum is based on the number of eigenvalues of the Hessian which are small. Minimizers with a large number of large eigenvalues tend to have worse generalization ability (Keskar et al., 2017). Let $\lambda_i(H)(\boldsymbol{w})$ be the $i^{th}$ largest eigenvalue of the Hessian of the loss function evaluated at $\boldsymbol{w}$. We can quantify the notion of sharpness by the number of eigenvalues larger than a threshold $\varepsilon \in \mathbb{R}^{>0}$:

$$\phi_1(\boldsymbol{w}, \varepsilon) = |\{\lambda_i(H)(\boldsymbol{w}) : \lambda_i > \varepsilon\}|. \tag{61}$$

A related sharpness metric uses the logarithm of the product of the $k$ largest eigenvalues (Wu et al., 2017),

$$\phi_2(\boldsymbol{w}, k) = \sum_{i=1}^{k} \log \lambda_i(H)(\boldsymbol{w}). \tag{62}$$

Both metrics require computing the eigenvalues of the Hessian. As a result, optimizing on these metrics during teleportation is prohibitively expensive. Hence, in this paper we use the average change in loss averaged over random directions ($\phi$) as objective in generalization experiments.

## E.2  MORE INTUITION ON CURVATURES AND GENERALIZATION

### E.2.1  EXAMPLE: CURVATURE AFFECTS AVERAGE DISPLACEMENT OF MINIMA

Consider an optimization problem with two variables $w_1, w_2 \in \mathbb{R}$. Assume that the minimum is a curve $\gamma : \mathbb{R} \to \mathbb{R}^2$ in the two-dimensional parameter space. For a point $\boldsymbol{w}_0$ on $\gamma$, we estimate its generalization ability by computing the expected distance between $\boldsymbol{w}_0$ and the new minimum obtained by shifting $\gamma$.

We consider the following two curves as examples:

$$\begin{aligned} \gamma_1 &: \mathbb{R} \to \mathbb{R}^2, t \mapsto (t, k_1 t^2) \\ \gamma_2 &: [0, 2\pi] \to \mathbb{R}^2, \theta \mapsto (k_2 \cos(\theta), k_2 \sin(\theta) + k_2), \end{aligned} \tag{63}$$

with $k_1, k_2 \in \mathbb{R}^{\neq 0}$. The curve $\gamma_1$ is a parabola with curvature $\kappa_1 = 2k_1$ at $\boldsymbol{w}_0 = (0, 0)$. The curve $\gamma_2$ is a circle, with curvature $\kappa_2 = \frac{1}{k_2}$ at $\boldsymbol{w}_0$. Note that $\gamma_1$ is the only polynomial approximation with integer power ($\gamma(t) = (t, k|t|^n), n \in \mathbb{Z}^+$) where the curvature at $\boldsymbol{w}_0$ depends on $k$. When $n < 1$, the value of $\boldsymbol{w}_0$ is undefined. When $n = 1$, the first derivative at $\boldsymbol{w}_0$ is undefined. When $n > 2$, $\kappa(\boldsymbol{w}_0) = 0$.

Assume that a distribution shift in data causes $\gamma$ to shift by a distance $r$, and that the direction of the shift is chosen uniformly at random over all possible directions. Viewing from the perspective of the curve, this is equivalent to shifting $\boldsymbol{w}_0$ by distance $r$.

The distance between a point $\boldsymbol{w}$ and a curve $\gamma$ is

$$dist(\boldsymbol{w}, \gamma) = \min_{\boldsymbol{w}' \in \gamma_2} \|\boldsymbol{w}' - \boldsymbol{w}\|_2. \tag{64}$$

Let $S_r$ be the circle centered at the origin with radius $r$. The expected distance between the old solution $\boldsymbol{w}_0$ and shifted curve is

$$\mathbb{E}_{\boldsymbol{w} \in S_r}[dist(\boldsymbol{w}, \gamma)] = \frac{\int_{S_r} dist(\boldsymbol{w}, \gamma) ds}{\int_{S_r} ds} = \frac{\int_0^{2\pi} dist((r\cos\theta, r\sin\theta), \gamma) r d\theta}{\int_0^{2\pi} r d\theta}. \tag{65}$$

In the limit of zero curvature, $\gamma$ is a straight line $\gamma(t) = (t, 0)$. In this case, the expected distance is

$$\mathbb{E}_{\boldsymbol{w} \in S_r}[dist(\boldsymbol{w}, \gamma)] = \frac{\int_0^{2\pi} |r\sin\theta| r d\theta}{2\pi r} = \frac{2r}{\pi} \approx 0.637 r. \tag{66}$$

Figure 7(b)(c) shows that the expected distance's dependence on $\kappa$. Using both curves $\gamma_1$ and $\gamma_2$, the generalization ability of $w_0$ depends on the curvature at $w_0$. However, the type of dependence is affected by the type of curve used. In other words, the curvatures at points around $w_0$ affect how the curvature at $w_0$ affects generalization. Therefore, from these results alone, one cannot deduce whether minima with sharper curvatures generalize better or worse. To find a more definitive relationship between curvature and generalization, further investigation on the type of curves on the minimum is required.

We emphasize that this example only serves as an intuition for connecting curvature to generalization. As a future direction, it would be interesting to consider different families of parametric curves, higher dimensional parameter spaces, and deforming in addition to shifting the minima.

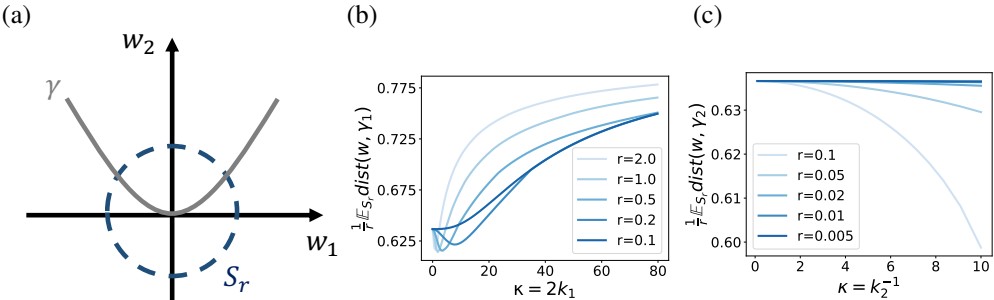

Figure 7: (a) Illustration of the parameter space, the minimum ($\gamma$), and all shifts with distance $r$ ($S_r$). (b) Expected distance between $w_0$ and the new minimum as a function of $\kappa$, for quadratic approximation $\gamma_1$. (c) Expected distance between $w_0$ and the new minimum as a function of $\kappa$, for constant curvature approximation $\gamma_2$. The expected distance is scaled by $r$ so that the curves can be plotted together.

### E.2.2 HIGHER DIMENSIONS

Figure 8 visualizes a curve obtained from a 2D minimum. However, it is not immediately clear what curves look like on a higher-dimensional minimum. A possible way to extend previous analysis is to consider sectional curvatures.

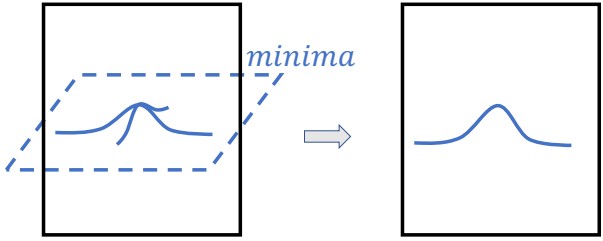

Figure 8: Left: a 2D minima in a 3D parameter space. Right: a 2D subspace of the parameter space and a curve on the minima (the intersection of the minima and the subspace).

### E.3 COMPUTING CORRELATION TO GENERALIZATION

We generate the 100 different models used in Section 4.3 by training randomly initialized models. For all three datasets (MNIST, FashionMNIST, and CIFAR-10), we train on 50,000 samples and test on a different set of 10,000 samples. The labels for classification tasks belongs to 1 of 10 classes.

For a batch of flattened input data $X \in \mathbb{R}^{d \times 20}$ and labels $Y \in \mathbb{R}^{20}$, the loss function is $\mathcal{L}(W_1, W_2, W_3, X, Y) = \text{CrossEntropy}\left(W_3 \sigma(W_2 \sigma(W_1 X)), Y\right)$, where $W_3 \in \mathbb{R}^{10 \times h_2}$, $W_2 \in \mathbb{R}^{h_2 \times h_1}$, $W_1 \in \mathbb{R}^{h_1 \times d}$ are the weight matrices, and $\sigma$ is the LeakyReLU activation with slope coefficient 0.1. For MNIST and Fashion-MNIST, $d = 28^2$, $h_1 = 16$, and $h_2 = 10$. For CIFAR-10,

$d = 32^3 \times 3$, $h_1 = 128$, and $h_2 = 32$. The learning rate for stochastic gradient descent is $0.01$ for MNIST and Fashion-MNIST, and $0.02$ for CIFAR-10. We train each model using mini-batches of size 20 for 40 epochs.

When computing the sharpness $\phi$, we choose the displacement list T that gives the highest correlation. The displacements used in this paper are $T = 0.001, 0.011, 0.021, ..., 0.191$ for MNIST, and $T = 0.001, 0.011, 0.021, ..., 0.191$ for Fashion-MNIST and CIFAR-10. We evaluate the change in loss over $|D| = 200$ random directions. For curvature $\psi$, we average over $k = 1$ curves generated by random Lie algebras (invertible matrices in this case).

Figure 9 and 10 visualizes the correlation result in Table 1. Each point represents one model.

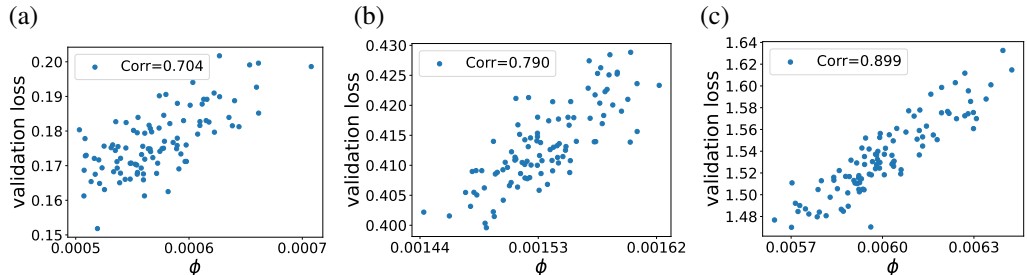

Figure 9: Correlation between sharpness and validation loss on MNIST (left), Fashion-MNIST (middle), and CIFAR-10 (right). Sharpness and generalization are strongly correlated.

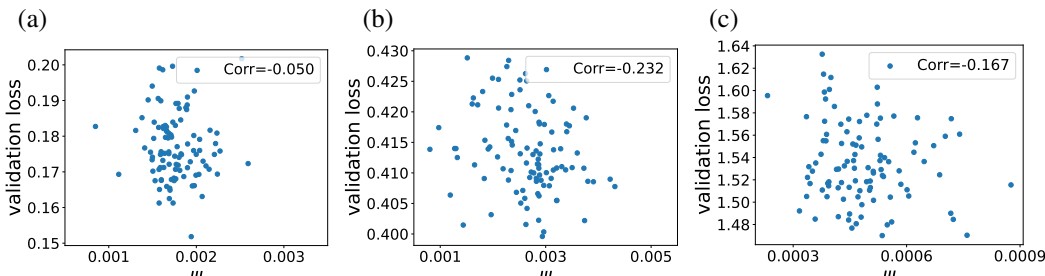

Figure 10: Correlation between curvature and validation loss on MNIST (left), Fashion-MNIST (middle), and CIFAR-10 (right). There is a weak negative correlation in all three datasets.

### E.4 ADDITIONAL DETAILS FOR GENERALIZATION EXPERIMENTS

Algorithm 2 shows an example on how to perform a teleportation with an MLP.

---

**Algorithm 2** Changing curvature using teleportation

---

**Input:** loss function $L(w)$, parameters before teleportation $w_0$, teleportation learning rate $\eta_{teleport}$, number of teleportation steps $n_{teleport}$.
**Output:** parameters after teleportation $w_{n_{teleport}}$.
**for** $t = 0$ **to** $n_{teleport} - 1$ **do**
    initialize $T = 0_{h \times h}$
    set $w'_t = (I_{h \times h} + T) \cdot w_t$
    compute $grad = \frac{d|\psi(w'_t)|}{dT}$
    set $T_t = \eta_{teleport} \times grad$
    set $w_{t+1} = (I + T_t) \cdot w_t$
**end for**
**Return** $w_{n_{teleport}}$

---

On CIFAR-10, we run SGD using the same three-layer architecture as in Section E.3, but with a smaller hidden size $h_1 = 32$ and $h_2 = 10$. At epoch 20 which is close to convergence, we teleport using 5 batches of data, each of size 2000. During each teleportation for $\phi$, we perform 10 gradient ascent (or descent) steps on the group element. During each teleportation for $\psi$, we perform 1 gradient ascent (or descent) step on the group element. The learning rate for the optimization on group elements is $5 \times 10^{-2}$.

To investigate how teleportation affects generalization for other optimizers, we repeat the same experiment but replace SGD with AdaGrad. Figure 11 shows the training curve of AdaGrad on CIFAR-10, averaged across 5 runs. Similar to SGD, changing curvature via teleportation affects the validation loss, while changing sharpness has negligible effects. Teleporting to points with larger curvatures helps find minima with slightly lower validation loss. Teleporting to points with smaller curvatures increases the gap between training and validation loss.

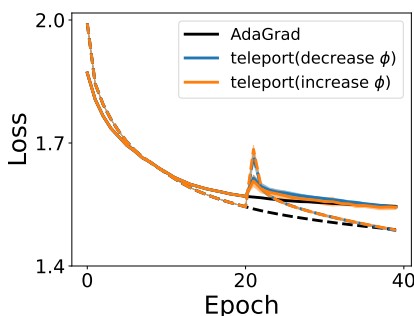 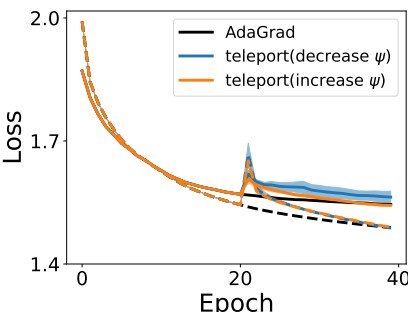

Figure 11: Changing sharpness (left) or curvature (right) using teleportation and its effect on generalizability of AdaGrad solutions on CIFAR-10. Solid line represents average test loss, and dashed line represent average training loss.

# F  INTEGRATING TELEPORTATION WITH OTHER GRADIENT-BASED ALGORITHMS

## F.1  DIFFERENT METHODS OF INTEGRATING TELEPORTATION WITH MOMENTUM AND ADAGRAD

**Setup.**  We test teleportation with various algorithms using the a 3-layer neural network and mean square error: $\min_{W_1,W_2,W_3} \|Y - W_3\sigma(W_2\sigma(W_1 X))\|_2$, with data $X \in \mathbb{R}^{5 \times 4}$, target $Y \in \mathbb{R}^{8 \times 4}$, and weight matrices $W_3 \in \mathbb{R}^{8 \times 7}$, $W_2 \in \mathbb{R}^{7 \times 6}$, and $W_1 \in \mathbb{R}^{6 \times 5}$. The activation function $\sigma$ is LeakyReLU with slope coefficient 0.1. Each element in the weight matrices is initialized uniformly at random over $[0, 1]$. Data $X, Y$ are randomly generated also from $[0, 1]$.

**Momentum.**  We compare three strategies of integrating teleportation with momentum: teleporting both parameters and momentum, teleporting parameters but not momentum, and reset momentum to 0 after a teleportation. In each run, we teleport once at epoch 5. Each strategy is repeated 5 times.

The training curves of teleporting momentum in different ways are similar (Figure 12a), possibly because the momentum accumulated is small compared to the gradient right after teleportations. All methods of teleporting momentum improves convergence, which means teleportation works well with momentum.

**AdaGrad.**  In AdaGrad, the rate of change in loss is

$$\frac{d\mathcal{L}(\boldsymbol{w})}{dt} = \frac{\partial \mathcal{L}}{\partial \boldsymbol{w}}^T \frac{d\boldsymbol{w}}{dt} = -\eta \|\nabla \mathcal{L}\|_A, \tag{67}$$

where $\eta \in \mathbb{R}$ is the learning rate, and $\|\nabla \mathcal{L}\|_A$ is the Mahalanobis norm with $A = (\varepsilon I + diag(G_{t+1}))^{-\frac{1}{2}}$. Previously, we optimize $\|\nabla \mathcal{L}\|_2$ in teleportation. We compare that to optimizing $\|\nabla \mathcal{L}\|_A$. Since the magnitude of $A$ is different than 1, a different learning rate for the gradient

ascent in teleportation is required. We choose the largest learning rate (with two significant figures) that does not lead to divergence. The teleportation learning rates used are $1.2 \times 10^{-5}$ for objective $\max_g \|\nabla \mathcal{L}\|_2$ and $7.5 \times 10^{-3}$ for objective $\max_g \|\nabla \mathcal{L}\|_A$.

Teleporting using the group element that optimizes $\|\nabla \mathcal{L}\|_A$ has a slight advantage (Figure 12b). Similar to the observations in Zhao et al. (2022), teleportation can be integrated into adaptive gradient descents.

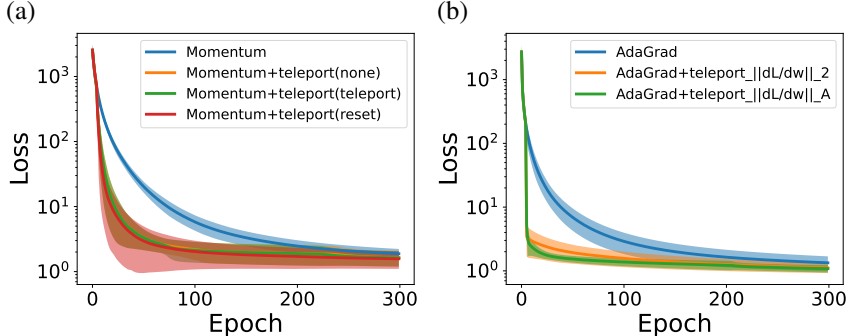

Figure 12: Comparison of different methods of integrating teleportation with momentum and Ada-Grad.

### F.2 ADDITIONAL DETAILS FOR EXPERIMENTS ON MNIST

We use a three-layer model and cross-entropy loss for classification with minibatches of size 20. For a batch of flattened input data $X \in \mathbb{R}^{28^2 \times 20}$ and labels $Y \in \mathbb{R}^{20}$, the loss function is $\mathcal{L}(W_1, W_2, W_3, X, Y) = \text{CrossEntropy}(W_3 \sigma(W_2 \sigma(W_1 X)), Y)$, where $W_3 \in \mathbb{R}^{10 \times 10}$, $W_2 \in \mathbb{R}^{10 \times 16}$, $W_1 \in \mathbb{R}^{16 \times 28^2}$ are the weight matrices, and $\sigma$ is the LeakyReLU activation with slope coefficient 0.1. The learning rates are $10^{-4}$ for AdaGrad, and $5 \times 10^{-2}$ for SGD with momentum, RMSProp, and Adam. The learning rate for optimizing the group element in teleportation is $5 \times 10^{-2}$, and we perform 10 gradient ascent steps when teleporting using each mini-batch. We use 50,000 samples from training set for training, and 10,000 samples in the test set for testing.

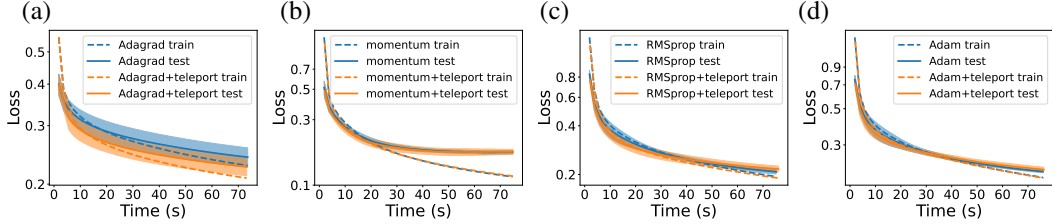

Figure 13: Runtime comparison for integrating teleportation into various algorithms. Solid line represents average training loss, and dashed line represents average test loss. Shaded areas are 1 standard deviation of the test loss across 5 runs. The plots look almost identical to Figure 5, indicating that the cost of teleportation is negligible compared to gradient descents.

