# OpenReview forum: "Improving Convergence and Generalization Using Parameter Symmetries"
_ICLR.cc/2024/Conference — ICLR 2024 oral_

### Official Review · Reviewer_dBnH · 2023-10-30

**Soundness:** 2 fair
**Presentation:** 2 fair
**Contribution:** 3 good
**Rating:** 6
**Confidence:** 3

**Summary:**

This paper analyzes the effect of teleportation technique in improving training and generalization performance of deep learning tasks. The authors first analyze the convergence property of teleportation in SGD and Newton's method. The effect of teleportation in improving generalization is justified using the concept of sharpness and curvature. Some numerical experiments verify the findings.

**Strengths:**

The paper proposes a convergence analysis for SGD with teleportation, and shows that it can improve the generalization of the solution. This demonstrates that teleportation has the potential to become a standard tool in modern deep learning training tasks.

**Weaknesses:**

1. My foremost concern is that, compared to existing work on parameter symmetry [1], this work seems more or less incremental in theory, basically extending the analysis from [1] to SGD and Newton's method. Moreover, the extension to SGD is done not based on standard noise assumption. The new results on generalization are justified mostly by experiments, rather than in theory.
2. The main assumptions are not clearly stated. I would suggest separating the assumptions rather than stating them at the beginning of each result. Moreover, the main result switches between stochastic (SGD) and deterministic settings (Newton's method), which makes it less accessible to the readers.

**Questions:**

1. What's the actual cost of one teleportation operation (group action) in practice? The authors conduct experiments on small-scale tasks. For huge deep networks, is it feasible to carry out one single teleportaion?
2. Is it possible to show that teleportation achieves better generalization bounds using tools like algorithm stability [2] ? The current results are encouraging but a bit lack of rigourous theoretical proof.

**References**

[1] Zhao, B., Dehmamy, N., Walters, R., & Yu, R. (2022). Symmetry teleportation for accelerated optimization. *Advances in Neural Information Processing Systems*, *35*, 16679-16690.

[2] Hardt, M., Recht, B., & Singer, Y. (2016, June). Train faster, generalize better: Stability of stochastic gradient descent. In *International conference on machine learning* (pp. 1225-1234). PMLR.

---

> ### Author Response · Authors · 2023-11-22
>
> Thank you for your comments!
>
> **Response to weaknesses**
>
> > compared to existing work on parameter symmetry [1], this work seems more or less incremental in theory, basically extending the analysis from [1] to SGD and Newton's method. Moreover, the extension to SGD is done not based on standard noise assumption.
>
> We respectfully disagree this is an incremental contribution in theory. Theorem 3.1 is the first global convergence theory that allows for teleportation. It was not trivial to identify that such a proof is possible, nor was it trivial to establish. We have also never seen a proof that guarantees that all points on the orbit converge before. There were three main technical challenges/innovations that set this theorem apart from a standard SGD theorem 1) finding the right proof structure and assumptions, 2) introducing a weighted telescoping, and  3) controlling an exponential terms with a cyclic dependency. Below we detail these three challenges.
> 1) Before proving Theorems 3.1, we first had to find a proof structure and assumptions that could work with teleportation. Standard proofs based on convexity, or strong convexity, were not amenable to a teleportation step. The reason being that the teleportation step can move far away from the current iterate, which in turn breaks the recurrence relations required by these proofs. We found that only non-convex type proofs could allow for teleportation.
>
> 2) After determining a smoothness type proof could work, one difficult point was to handle terms that do not telescope. That is, if you look to eq (28) in the supp. material, the suboptimality terms $(L(w^t)-L(w^*))$ and $(L(w^{t+1})-L(w^*))$ do not telescope because they have different constants weighting these terms. To resolve this, we introduced artificial weights $\alpha_t$ so that the suboptimality terms in eq (29) would telescope. This step is very different from a regular SGD proof.
>
> 3) After telescoping eq (29), the weighted telescoping generated terms that grew exponentially, see the term in eq (33). We had to control the growth of this exponential term by making $T$ big enough, but this created a circular dependency between $T$ and the step size $\eta$. Fortunately we were able to resolve this dependency, and still arrive at a. $1/\sqrt{T}$ complexity result.
>
>
> Regarding the noise assumption, note that our theorem makes no assumption on the noise, which makes it more general than theorems that suppose a certain noise structure. The more classical SGD proofs assume that the stochastic gradients are bounded (equivalently the loss if Lipschitz), that is ||g_t|| <= D which is a fixed constant. We make no explicit assumption on the norm of the gradient, instead, we bound E||g_t|| as a result of assuming the loss is a smooth function, see Lemma A.1 in our paper. This Lemma A.1 and approach for bounding the gradient is standard when assuming smoothness.
>
> > The new results on generalization are justified mostly by experiments, rather than in theory.
>
> While the main text contains mostly experimental results, we discuss a possible theoretical approach in Appendix F.2. In short, we consider the shift of the minimum under data distribution shifts, and examine how curvature affects the average distance between the old minimum and the new minimum. The correlation depends on the shape of the minimum and may be more complicated beyond a two-dimensional parameter space. We leave a more systematic investigation to future work.
>
> How sharpness affects generalization has been studied extensively but not yet fully understood [3]. We expect the theory on the curvature of minimum to be comparably complex and intriguing.
>
> > The main assumptions are not clearly stated. I would suggest separating the assumptions rather than stating them at the beginning of each result. Moreover, the main result switches between stochastic (SGD) and deterministic settings (Newton's method), which makes it less accessible to the readers.
>
> We have added a clarification on the settings in Section 3.2. We plan to add a section dedicated to preliminaries and assumptions in the final version of the paper.

---

> ### Author Response · Authors · 2023-11-22
>
> **Response to Questions**
> > “1. What's the actual cost of one teleportation operation (group action) in practice? The authors conduct experiments on small-scale tasks. For huge deep networks, is it feasible to carry out one single teleportation?”
>
> The complexity for one gradient ascent step in teleportation is the same as one step of back-propagation [1]. The time required to teleport a pair of layers scales quadratically with the hidden layer width and linearly with the mini-batch size. For large models, one can teleport a subset of all weights without affecting the others, as individual teleportation can be performed on each pair of consecutive layers. Since the time complexity of teleporting a pair of layers does not depend on the depth, it is feasible to carry out teleportation in deep networks.
>
> > “2. Is it possible to show that teleportation achieves better generalization bounds using tools like algorithm stability [2]? The current results are encouraging but a bit lack of rigourous theoretical proof.”
>
> Thanks for pointing us to a relevant approach for developing a generalization bound. The generalization bounds in [2] build on the theorem that uniform stability implies generalization in expectation, and that stochastic gradient methods are $\epsilon$-uniformly stable. One approach to show that teleportation achieves better generalization bounds is to prove that SGD with a teleportation that improves curvature is $\epsilon_1$-uniformly stable with $\epsilon_1 < \epsilon$, which leads to a tighter generalization bound. However, establishing such a proof appears to be challenging, since it is not clear how teleportation affects uniform stability.
>
> Our main contribution in Section 4 is developing a new approximation method for the curvature of the minimum, and effect of teleporting for sharpness/curvature on generalization. Finding a complete theory is beyond the scope of this paper. However, we have included some theoretical investigations on the correlation between curvature and generalization. We believe that Appendix F.2 could serve as a starting point to develop a generalization bound for teleportation. The expected displacement of minimum under distribution shift is directly related to the curvature of the minimum and provides a way to quantify the generalization gap.
>
> **References**
>
> [1] Zhao, B., Dehmamy, N., Walters, R., & Yu, R. (2022). Symmetry teleportation for accelerated optimization. Advances in Neural Information Processing Systems, 35, 16679-16690.
>
> [2] Hardt, M., Recht, B., & Singer, Y. (2016, June). Train faster, generalize better: Stability of stochastic gradient descent. In International conference on machine learning (pp. 1225-1234). PMLR.
>
> [3] Maksym Andriushchenko, Francesco Croce, Maximilian Müller, Matthias Hein, Nicolas Flammarion. A Modern Look at the Relationship between Sharpness and Generalization. International Conference on Machine Learning, 2023.

---

> > ### Comment · Reviewer_dBnH · 2023-11-22
> > **Thank you for your response**
> >
> > Thank you for your response. The reviewer addressed most of my concerns regarding the scalability of the proposed approach.
> > It is recommended if you could add some experiments on deep networks and evaluate effect of teleportation on training time.
> > Overall I raise my score to 6.

---

> ### Author Response · Authors · 2023-11-23
>
> Thank you for the suggestion. We will validate the theoretical time complexity by evaluating teleportation's runtime in larger scale experiments in future work.

---

### Official Review · Reviewer_Ewvi · 2023-11-01

**Soundness:** 3 good
**Presentation:** 4 excellent
**Contribution:** 3 good
**Rating:** 8
**Confidence:** 3

**Summary:**

Teleportation is the transformation of the parameters in the parameters such that the loss is unchanged. This work theoretically shows that teleportation has accelerating effects on the convergence rate of SGD for non-convex loss. Furthermore, through experiments, they show that teleportation can also improve generalization by moving the parameters to a flatter region.

**Strengths:**

- The authors are able to prove a stronger convergence guarantee for SGD by using transportation. Instead of having a guarantee for a single stationary point like the classic SGD, they show that SGD with teleportation has convergence guarantees for a set of stationary points in group $G$. The intuition on why SGD with teleportation has accelerating effects is well explained by connecting it with second-order algorithms.

- The paper shows that one teleportation might be enough which makes it feasible to do in practice.

- The curvature of minima seems like a pretty interesting way to understand the generalization ability of the stationary points.

- The paper is well-written overall.

**Weaknesses:**

- I'm a bit confused about the claim that teleportation accelerates the convergence rate of SGD. The intuition part makes sense to me since it has the quadratic error term that typically arises from second-order optimization but the convergence rate in Theorem 3.1 is still $O(\epsilon^{-4}$ ), which is the same as SGD. The convergence guarantee is slightly stronger but I don't understand how we can claim teleportation accelerates the convergence rate of SGD.

- Even though the claim is teleportation improves the convergence rate for Adagrad, SGD with momentum, RMSProp, and Adam, the only clear improvements that I could see from Figure 5 is Adagrad. The other graphs seem to have similar performance for algorithm with or without teleportation.

**Questions:**

- Does cross-entropy loss satisfy the condition for one teleportation in section 3.3? The experiment on Cifar 10 uses one-time teleportation but the remark in section 3.3 only mentions quadratic loss.

- Does it matter at which epoch we perform the teleportation? If it does, how do we pick the best epoch?

- In page 7, it says teleporting to points with smaller curvature helps find a minimum with lower validation loss but in the left graph of figure 4, the loss when we move to a place with decreased curvature is actually higher. Am I missing something here?

---

> ### Author Response · Authors · 2023-11-22
>
> Thank you for your comments and positive feedback!
>
> **Response to weaknesses**
>
> > I'm a bit confused about the claim that teleportation accelerates the convergence rate of SGD. The intuition part makes sense to me since it has the quadratic error term that typically arises from second-order optimization but the convergence rate in Theorem 3.1 is still O(\epsilon^{-4}), which is the same as SGD. The convergence guarantee is slightly stronger but I don't understand how we can claim teleportation accelerates the convergence rate of SGD.
>
> This is a good question, and we agree that “accelerates” is not the correct word here. Instead, we give a stronger convergence guarantee by showing that all parameters on the orbit of the iterates converge to a stationary point at a O(\epsilon^{-4}) rate, as opposed to the standard SGD theory which shows only the iterates converge at this rate. Thus the speed of convergence is the same, but the object that is converging (all points on the orbit) is a stronger notion of convergence.  We will re-word our contribution here accordingly, and thank the reviewer.
>
> > Even though the claim is teleportation improves the convergence rate for Adagrad, SGD with momentum, RMSProp, and Adam, the only clear improvements that I could see from Figure 5 is Adagrad. The other graphs seem to have similar performance for algorithm with or without teleportation.
>
> We agree that the improvement brought by teleportation is less significant for algorithms other than SGD, perhaps due to how they handle gradients of different magnitudes. However, for all algorithms, the loss curve drops visibly faster right after teleportation, near the beginning of the training. Since teleportation does not have a significant impact on total runtime (last Figure in Appendix), we believe that it is still beneficial to integrate it with various optimizers.
>
> **Response to Questions**
> > “Does cross-entropy loss satisfy the condition for one teleportation in section 3.3? The experiment on Cifar 10 uses one-time teleportation but the remark in section 3.3 only mentions quadratic loss.”
>
> It is difficult to analytically check whether cross-entropy loss satisfies the condition in section 3.3, although we observed in the experiment that one teleportation is usually sufficient to improve the convergence speed.
>
> > “Does it matter at which epoch we perform the teleportation? If it does, how do we pick the best epoch?”
>
> We did not conduct a systematic investigation of when to teleport in this paper, but results in a previous paper [1] suggests that for the purpose of improving convergence, teleporting around the beginning of training leads to the most significant improvement. For improving generalization, we teleport when SGD is close to being converged, because the sharpness and curvature metrics are designed for when loss is near 0. Additionally, these quantities can change during training due to the implicit bias of optimization algorithms, so optimizing them early in the training may not be effective.
>
> > “In page 7, it says teleporting to points with smaller curvature helps find a minimum with lower validation loss but in the left graph of figure 4, the loss when we move to a place with decreased curvature is actually higher. Am I missing something here?”
>
> This is a typo in the text and has been fixed. Thanks for pointing it out!

---

### Official Review · Reviewer_BdD6 · 2023-11-02

**Soundness:** 4 excellent
**Presentation:** 4 excellent
**Contribution:** 2 fair
**Rating:** 8
**Confidence:** 3

**Summary:**

Typical deep learning models have natural symmetries on their weight parameters that leave their output unchanged. The study of parameter symmetries in deep learning focuses on the effects of these symmetries in terms of optimization and generalization performance. The present work contributes to this burgeoning field by first offering an analysis of "teleported" SGD, an algorithm proposed by [1] which uses the symmetries to move the current iterate to one with largest (under the group action) gradient. Their first contribution is an analysis which shows that teleported SGD yields improved guarantees for SGD for smooth (possibly non-convex) functions. It was shown in [1] that teleportation (in the GD rather than SGD setting) is equivalent to a step of Newton's method, and they accordingly give an estimate of its contraction rate that demonstrates quadratic convergence as would be expected from a 2nd order method. Their final theoretical results are sufficient conditions for one teleportation step to be optimal for all times. They then provide empirical studies of the effects of teleportation for increasing/decreasing the sharpness and curvature of minima. Finally, the authors consider teleported variants of other standard optimization algorithms (Adagrad, SGD with momentenum, RMS prop, and Adam), and propose a method for meta-learning the teleportation.

[1] Symmetry Teleportation for Accelerated Optimization, by Zhao, Dehmamy, Walters, and Yu 2022

**Strengths:**

This paper is written in a very clear and engaging style, and was a pleasure to read. The exploration of sufficient conditions for one-teleportation to be enough (section 3.3, especially Prop. 3.4) and the effects of teleporting for sharpness/curvature (section 4), plus their computationally feasible proxies $\phi$ and $\psi$ are, to my understanding, significant and novel. Finally, their empirical results are clear and original. For example, Figure 5 on the effects of teleportation on other first-order algorithms is particularly convincing. And their use of Pearson correlation in Table 1 to estimate the effects of curvature and sharpness is impressive.

**Weaknesses:**

Overall, I really like the paper. One weakness, however, is that some of their theoretical results are not especially novel. For example their results on teleported SGD (Theorem 3.1) and the Newton steps (Prop. 3.2) seems to be minor modifications of standard proof techniques. And the (very interesting) fact that teleported SGD is equivalent to a Newton iteration was already observed by previous work [1]. However, this is no way inclines me to reject the paper -- I think its clarity and other contributions are more than enough to merit acceptance.

[1] Symmetry Teleportation for Accelerated Optimization, by Zhao, Dehmamy, Walters, and Yu 2022

**Questions:**

- It seems that Definition 3.3 doesn't exclude *minimizers* of $w \mapsto \|\partial \mathcal{L}/\partial w\|^2$. It might help the exposition to mention this.
- I am a bit confused about Figure 4: the paper text says "teleporting to points with smaller curvatures helps find a minimum with lower validation loss, while teleporting to points with larger curvatures has the opposite effect". And this relationship, that less curvature is associated with better generalization performance, is also present in Table 1. However, Figure 4 seems to show the opposite phenomenon: the teleport(increase $\psi$) test loss is better than the teleport(decrease $\psi$) test loss. Am I mis-reading this plot? Could you please clarify this point, ideally in the paper text as well?
- "from Lemma A.1 below" before equation 26 and on page 13 should instead read "from Lemma A.1 above"

---

> ### Author Response · Authors · 2023-11-22
>
> Thank you for your comments and positive feedback!
>
> We respectfully disagree that the theoretical results in the first part of Section 3 are not novel. Theorem 3.1 is the first global convergence theory that allows for teleportation. It was not trivial to identify that such a proof is possible, nor was it trivial to establish. There were three main technical challenges/innovations that set this theorem apart from a standard SGD theorem 1) finding the right proof structure and assumptions, 2) introducing a weighted telescoping, and  3) controlling an exponential terms with a cyclic dependency. Below we detail these three challenges.
>
> 1) Before proving Theorems 3.1, we first had to find a proof structure and assumptions that could work with teleportation. Standard proofs based on convexity, or strong convexity, were not amenable to a teleportation step. The reason being that the teleportation step can move far away from the current iterate, which in turn breaks the recurrence relations required by these proofs. We found that only non-convex type proofs could allow for teleportation.
>
> 2) After determining a smoothness type proof could work, one difficult point was to handle terms that do not telescope. That is, if you look to eq (28) in the supp. material, the suboptimality terms $(L(w^t)-L(w^*))$ and $(L(w^{t+1})-L(w^*))$ do not telescope because they have different constants weighting these terms. To resolve this, we introduced artificial weights $\alpha_t$ so that the suboptimality terms in eq (29) would telescope. This step is very different from a regular SGD proof.
>
> 3) After telescoping eq (29), the weighted telescoping generated terms that grew exponentially, see the term in eq (33). We had to control the growth of this exponential term by making $T$ big enough, but this created a circular dependency between $T$ and the step size $\eta$. Fortunately we were able to resolve this dependency, and still arrive at a. $1/\sqrt{T}$ complexity result.
>
>
> **Response to Questions**
> > “It seems that Definition 3.3 doesn't exclude minimizers of $w \mapsto |\partial \mathcal{L}/\partial w|^2$. It might help the exposition to mention this.”
>
> Thanks for the suggestion! We agree that this point is not clear in the original draft and have made this explicit in the revised version.
>
> > Figure 4 seems to show the opposite phenomenon to text description.
>
> This is a typo in the text and has been fixed. Figure 4 shows that teleporting to points with *larger* curvatures helps find a minimum with lower validation loss, while teleporting to points with *smaller* curvatures has the opposite effect. This relationship is consistent with Table 1. For the dataset and architecture we examined, less curvature is associated with larger validation loss, which is associated with worse generalization.

---

> ### Comment · Reviewer_BdD6 · 2023-11-22
> **Thanks for your response**
>
> Thanks for your thoughtful response. I wanted to add that I don't think the re-weighting trick in the proof of Theorem 1 is necessary. In particular, if you take $\eta_t = 1/\sqrt{T}$, then equation (29) becomes
> $$
> \frac{1}{\sqrt{T}} \mathbb{E}[\|\nabla \mathcal{L}(g^t \cdot w^t)\|^2] \leqslant
> \mathbb{E}[ \mathcal{L}(w^t) - \mathcal{L}(w^{t + 1})] + \frac{\beta^2}{T}(\mathbb{E}[\mathcal{L}(w^{t}) - \mathcal{L}\_\star] +
> \sigma^2).
> $$ So we find
> $$
> \min\_{t = 0, \ldots, T - 1} \mathbb{E}[ \| \nabla \mathcal{L}(g^t \cdot w^t) \|^2]
> \leqslant \frac{1}{T} \sum_{t = 1}^T \mathbb{E}[\|\nabla \mathcal{L}(g^t \cdot w^t)\|^2]
> \leqslant \frac{1}{\sqrt{T}} \mathbb{E}[ \mathcal{L}(w^1) - \mathcal{L}(w^{T})] + \frac{\beta^2 }{\sqrt{T}}
> (\mathbb{E}[ \mathcal{L}(w^1) - \mathcal{L}(w\_\star)] + \sigma^2).
> $$ Then since $\mathcal{L}(w^{T}) \geqslant \mathcal{L}\_\star$, we essentially recover Theorem 1, up to dependence on $\beta$ (and I think the exact statement of Theorem 1 is recovered by taking $\eta_t = 1/\beta\sqrt{T - 1}$ as in Theorem 1 itself).
>
> So from what I can tell, the proof is nearly done once equation (29) is obtained, and in particular there is no need for the re-weighting trick.
>
> I am leaving my score unchanged, and stand my by original assessment that this work clearly merits acceptance.

---

> > ### Author Response · Authors · 2023-11-22
> > **Missing a non-telescoped term**
> >
> > Dear Reviewer, thank you very much for engaging on the details.
> >
> > You're approach is interesting, but I see an issue. I agree that taking  $\eta_t = 1/\sqrt{T}$ in equation (29) gives
> > $$
> > \frac{1}{\sqrt{T}} \mathbb{E}[|\nabla \mathcal{L}(g^t \cdot w^t)|^2] \leqslant
> > \mathbb{E}[ \mathcal{L}(w^t) - \mathcal{L}(w^{t +1})]  + \frac{\beta^2}{T}(\mathbb{E}[\mathcal{L}(w^{t}) - \mathcal{L}_*]+\sigma^2)
> > $$
> >
> > But the next step doesn't quite work before this remaining $\mathbb{E}[\mathcal{L}(w^{t}) - \mathcal{L}_\star] $ term does not telescope with any other term. That is, from the above we get
> >
> > $$
> > \min_{t = 0, \ldots, T - 1} \mathbb{E}[ | \nabla \mathcal{L}(g^t \cdot w^t) |^2]
> > \leq \frac{1}{T} \sum_{t = 1}^T \mathbb{E}[|\nabla \mathcal{L}(g^t \cdot w^t)|^2]
> > \leq \frac{1}{\sqrt{T}} \mathbb{E}[ \mathcal{L}(w^1) - \mathcal{L}(w^{T})] + \frac{\beta^2 }{T^{3/2}} \sum_{t=1}^T
> >  \mathbb{E}[ \mathcal{L}(w^t) - \mathcal{L}(w_\star)] + \frac{\beta^2 }{\sqrt{T}}\sigma^2.
> > $$
> > Now the issue is how to control the $\sum_{t=1}^T
> >  \mathbb{E}[ \mathcal{L}(w^t) - \mathcal{L}(w_\star)]$ term that did not telescope, that was missing from your second equation.

---

> > > ### Comment · Reviewer_BdD6 · 2023-11-22
> > > **Thanks for your response**
> > >
> > > I was assuming that the terms $\mathbb{E}[\mathcal{L}(w^t)]$ are non-increasing in $t$, which would imply that $\mathbb{E}[\mathcal{L}(w^t) - \mathcal{L}\_\star] \leqslant \mathbb{E}[\mathcal{L}(w^1) - \mathcal{L}\_\star]$, and we would be able to conclude the result. However, it doesn't seem that this follows.
> > >
> > > This does raise a question though, which is that something like this would work if the noise term $\sigma^2$ were defined to be
> > > $$
> > > \tilde\sigma^2 := \sup\_{w} \mathbb{E}[\|\nabla \mathcal{L}(w) - \nabla \mathcal{L}(w, \xi)\|^2].
> > > $$ Since you could then rewrite equation (26) as
> > > $$
> > > \mathbb{E}\_t[\mathcal{L}(w^{t + 1})] \leqslant \mathcal{L}(w^t) - \eta\_t \|\nabla \mathcal{L}(g^t \cdot w^t)\|^2 + \frac{\beta \eta\_t^2}{2} \mathbb{E}\_t [\| \nabla \mathcal{L}(w, \xi)\|^2] \leqslant
> > >  \mathcal{L}(w^t) - (\eta\_t - \beta \eta\_t^2/2) \|\nabla \mathcal{L}(g^t \cdot w^t)\|^2 + \frac{\beta \eta\_t^2 \tilde \sigma^2}{2}
> > > $$ So that as long as $\eta\_t \leqslant 2/\beta$, we have
> > > $$
> > > \mathbb{E}\_t[\mathcal{L}(w^{t + 1})] \leqslant
> > >  \mathcal{L}(w^t) + \frac{\beta \eta\_t^2 \tilde \sigma^2}{2},
> > > $$ which finally means
> > > $$
> > > \mathbb{E}[\mathcal{L}(w^{t + 1})] \leqslant
> > >  \mathcal{L}(w^1) + \frac{\beta \tilde \sigma^2}{2} \sum_{k = 1}^t \eta_t^2.
> > > $$ So that if $\eta_t = 1/\beta\sqrt{T}$, then we would get
> > > $$
> > > \mathbb{E}[\mathcal{L}(w^{t + 1})] \leqslant
> > >  \mathcal{L}(w^1) + \frac{\tilde \sigma^2}{2},
> > > $$ for all $t$. We could then apply this to the term you mentioned and get a similar result as Theorem 1 but with the $\sigma^2$ replaced by $\tilde \sigma^2$ (you would also have to use a descent lemma with this $\tilde \sigma^2$ term). Anyways, this approach would require a change in your results so is not a simplification anymore. But it raises the question - what is the significance of your $\sigma^2$ term? Does this term appear in other works on non-convex optimization with SGD? It might be worth commenting on this in the main text as well.

---

> ### Author Response · Authors · 2023-11-22
> **The interpolation constant vs the global Lipschitz constant**
>
> I understood your approach and it is also interesting. But note that the noise constant you introduced requires assuming that the variance of the gradient estimate is globally bounded, and this is an additional assumption. We choose not to have this additional assumption, because it does not lead to an improvement in the rates (despite being more restrictive).
>
> With respect to the two different noise constants. Your noise constant $\tilde{\sigma}^2$  is more common in the convex and non-smooth setting, where we often assume the loss function is Lipschitz. Indeed if $\mathcal{L}(w,\xi)$ is Lipschitz in $w$, then your $\tilde{\sigma}^2$ is bounded. The noise constant $\sigma^2$ we use is sometimes called the interpolation constant, and it is more common in the smooth setting. It's called the interpolation constant because if the model fits perfectly the data than $\sigma^2 =0.$ Thus $\sigma^2$ measure how close we are to interpolation. This constant has recently appeared in other work in the non-convex setting. We will take the reviewers suggestion and add more references to this constant and explain it's meaning. Thanks again.

---

> > ### Comment · Reviewer_BdD6 · 2023-11-22
> > **Thanks for indulging me**
> >
> > Thanks very much for indulging me, I appreciate your detailed consideration and am happy that you will discuss this point a little more in the revised version.

---

### Official Review · Reviewer_bsXX · 2023-11-06

**Soundness:** 3 good
**Presentation:** 2 fair
**Contribution:** 4 excellent
**Rating:** 8
**Confidence:** 2

**Summary:**

The paper investigates how teleportation, i.e., applying a loss-invariant group action to the parameter space can improve (i) optimization speed and (ii) generalization in deep learning. For (i), the paper derives an upper bound for the gradient norm, which implies that SGD iterates converge to a basin of stationary points, from which only other stationary points are reachable via teleportation. They further show that SGD with teleportation can behave similar to Newton's method and provide a necessary condition on when one teleportation step is sufficent to accelerate optimization. They extend teleportation to commonly used optimizers and experimentally show that a teleportation step in the first epoch improves the convergence rate. Lastly, they incorporate teleportation into a meta-learned optimizer and show that learning the group element in teleportation improves the convergence rate of (meta-learned) gradient descent. For (ii), the paper introduces a novel measure for generalization based on the curvature of minima, and empirically shows that teleporting to points which decrease sharpness and increase the curvature of minima correlates with an improvement in generalization.

**Strengths:**

The paper provides novel results on exploiting parameter symmetries in the context of optimization and generalization in deep learning. For optimization, the theoretical results improve on existing work, while the paper appears to be the first to investigate teleportation with respect to generalization. The presented results also have promise to be of practical relevance, since the computational overhead of the teleportation step appears to be negligible in the experiments.

**Weaknesses:**

* **Clarity**: Although the paper is generally well-written, I did find it difficult to follow at times, especially with respect to the overall structure. One suggestion would be to switch the order of sections 4 and 5, as section 5 investigates how teleportation improves optimization, while section 4 is more or less self-contained with respect to generalization. I would also suggest having a (sub)section which is dedicated to introducing the necessary preliminaries and assumptions, with additional pointers to literature (e.g., some of the notation in section 3.3 could be introduced in the preliminaries already). In the appendix, it would be helpful to restate all the needed notation and equation, so the reader does not have to switch between the main paper and the appendix to follow the proofs.
* **Reproducibility**: If I am not mistaken, there is no reference to or mention of any source code; it would be great to make your code publicly available.

Please find some minor remarks below:

* p. 3: we provide theoretical analysis of teleportation -> we provide a theoretical analysis of teleportation
* p. 3: that maximizes the magnitude of gradient -> that maximizes the magnitude of the gradient
* p. 3: the iterates equation 4 -> the iterates in equation 4
* p. 3, Theorem 3.1: I assume $\theta$ should be $w$
* Proposition 3.2: I assume $f$ should be $\mathcal{L}$
* p. 5: To simplify notations -> To simplify notation
* p. 7: at the 20 epoch -> at the 20th epoch/at epoch 20
* p. 7: teleporting to sharper point -> teleporting to sharper points
* Lemma A.1: eq. (19) LHS: $\xi$ seems to be missing in $\mathcal{L}$, also 2 lines below

**Questions:**

1. In the experiments, have you tried teleporting more often than just in the first epoch and whether it has a positive effect on convergence speed/generalization vs. the increase in runtime?
2. How did you decide on 10 and 1 gradient ascent steps, respectively, for the experiments?
3. Do you have any (preliminary) results on how teleportation affects generalization for other optimizers (e.g., AdaGrad, Adam, etc.)?
4. This question goes beyond the scope of the paper, but I would be interested in your opinion on [1] in light of your contribution, a recent paper which challenges the current view on the correlation between sharpness and generalization.

[1] https://openreview.net/pdf?id=VZp9X410D3

---

> ### Author Response · Authors · 2023-11-22
>
> Thank you for the detailed comments. In the revised version, we have fixed the writing issues according to the list of minor remarks.
>
> **Clarity**
>
> We appreciate the suggestions on the overall structure. Section 5 is placed at the end since while experiments in that section focus on optimization, teleportation can be integrated with other optimizers both to improve optimization and to improve generalization. We agree that a section dedicated to preliminaries and assumptions would be helpful, but currently it is difficult to add to the main text due to the page limit. We will add more background in the final version of the paper. We have added the definition of needed notations in the appendix in the revised version.
>
> **Reproducibility**
>
> We have uploaded our source code together with the revised paper. We will add a github link in the final version of the paper.
>
> **Response to Questions**
> > “1. In the experiments, have you tried teleporting more often than just in the first epoch and whether it has a positive effect on convergence speed/generalization vs. the increase in runtime?”
>
> For improving convergence, the runtime added by teleportation is negligible, but we observed that teleporting after the first epoch has almost no effect on convergence speed, which is consistent with the results in [2] (section 6.2). For improving generalization, performing a teleportation to change curvature is more time consuming. Under our current implementation, the extra cost does not justify teleporting more than once near a minimum.
>
> > “2. How did you decide on 10 and 1 gradient ascent steps, respectively, for the experiments?”
>
> The number of gradient ascent steps is chosen such that the curvature or sharpness shows clear improvement but does not become large enough to cause divergence in the subsequent gradient descent steps.
>
> > “3. Do you have any (preliminary) results on how teleportation affects generalization for other optimizers (e.g., AdaGrad, Adam, etc.)?”
>
> Thank you for the suggestion! We added a new experiment that investigates how teleportation affects generalization for AdaGrad (Figure 12 in the revised paper). We found that similar to SGD, changing curvature via teleportation affects the validation loss, while changing sharpness has negligible effects. Teleporting to points with larger curvatures helps find minima with slightly lower validation loss. Teleporting to points with smaller curvatures increases the gap between training and validation loss.
>
> > “4. This question goes beyond the scope of the paper, but I would be interested in your opinion on [1] in light of your contribution, a recent paper which challenges the current view on the correlation between sharpness and generalization.”
>
> The paper by Andriushchenko et. al provides an extensive empirical study on the correlation between sharpness and generalization for transformers and ConvNets on large datasets. Their observations that sharpness does not correlate well with generalization and that the right sharpness measure is data-dependent reveals the complexity of the role of sharpness in generalization. Their results may also explain our observation that teleporting in symmetry directions to change sharpness has negligible effect on generalization. We expect curvature to have a comparably complex and intriguing effect on generalization. We believe that the link between sharpness/curvature and generalization is interesting and deserves further investigation, and we hope that teleportation could become useful as a tool to explore the minimum manifold.
>
> [1] Maksym Andriushchenko, Francesco Croce, Maximilian Müller, Matthias Hein, Nicolas Flammarion. A Modern Look at the Relationship between Sharpness and Generalization. International Conference on Machine Learning, 2023.
>
> [2] Bo Zhao, Nima Dehmamy, Robin Walters, Rose Yu. Symmetry teleportation for accelerated optimization. Advances in Neural Information Processing Systems, 2022.

---

> ### Comment · Reviewer_bsXX · 2023-11-22
>
> Thanks for the response and uploading your code!

---

### Meta-Review · Area_Chair_t965 · 2023-12-03

**Metareview:**

The reviewers are unanimous that this is a good submission and that it should be accepted. I encourage the authors to take into account the valuable comments from the reviewers in their camera-ready version.

**Justification For Why Not Higher Score:**

There is no higher score

**Justification For Why Not Lower Score:**

I have very much enjoyed this submission, and found it to be very interesting. The reviewers all agree that it should be accepted.

---

### Decision · Program_Chairs · 2024-01-16

Accept (oral)